# Incorporating Symmetry into Deep Dynamics Models for Improved Generalization

**Rui Wang** *
Computer Science and Engineering
University of California
San Diego, CA 92093
`ruw020@ucsd.edu`

**Robin Walters** *
Khoury College of Computer Science
Northeastern University
Boston, MA 02115
`r.walters@northeastern.edu`

**Rose Yu**
Computer Science and Engineering
University of California
San Diego, CA 92093
`roseyu@ucsd.edu`

## Abstract

Recent work has shown deep learning can accelerate the prediction of physical dynamics relative to numerical solvers. However, limited physical accuracy and an inability to generalize under distributional shift limits its applicability to the real world. We propose to improve accuracy and generalization by incorporating symmetries into convolutional neural networks. Specifically, we employ a variety of methods each tailored to enforce a different symmetry. Our models are both theoretically and experimentally robust to distributional shift by symmetry group transformations and enjoy favorable sample complexity. We demonstrate the advantage of our approach on a variety of physical dynamics including Rayleigh–Bénard convection and real-world ocean currents and temperatures. Compared with image or text applications, our work is a significant step towards applying equivariant neural networks to high-dimensional systems with complex dynamics. We open-source our simulation, data and code at `https://github.com/Rose-STL-Lab/Equivariant-Net`.

## 1 Introduction

Modeling dynamical systems in order to forecast the future is of critical importance in a wide range of fields including, e.g., fluid dynamics, epidemiology, economics, and neuroscience [2; 21; 45; 22; 14]. Many dynamical systems are described by systems of non-linear differential equations that are difficult to simulate numerically. Accurate numerical computation thus requires long run times and manual engineering in each application.

Recently, there has been much work applying deep learning to accelerate solving differential equations [46; 6]. However, current approaches struggle with generalization. The underlying problem is that physical data has no canonical frame of reference to use for data normalization. For example, it is not clear how to rotate samples of fluid flow such that they share a common orientation. Thus real-world out-of-distribution test data is difficult to align with training data. Another limitation of current approaches is low physical accuracy. Even when mean error is low, errors are often spatially correlated, producing a different energy distribution from the ground truth.

We propose to improve the generalization and physical accuracy of deep learning models for physical dynamics by incorporating symmetries into the forecasting model. In physics, Noether's Law gives a correspondence between conserved quantities and groups of symmetries. By building a neural network which inherently respects a given symmetry, we thus make conservation of the associated quantity more likely and consequently the model's prediction more physically accurate.

---

*Equal contribution

A function $f$ is equivariant if when its input $x$ is transformed by a symmetry group $g$, the output is transformed by the same symmetry,

$$f(g \cdot x) = g \cdot f(x).$$

See Figure 1 for an illustration. In the setting of forecasting, $f$ approximates the underlying dynamical system. The set of valid transformations $g$ is called the symmetry group of the system.

By designing a model that is inherently equivariant to transformations of its input, we can guarantee that our model generalizes automatically across these transformations, making it robust to distributional shift. The symmetries we consider, translation, rotation, uniform motion, and scale, have different properties, and thus we tailor our methods for incorporating each symmetry.

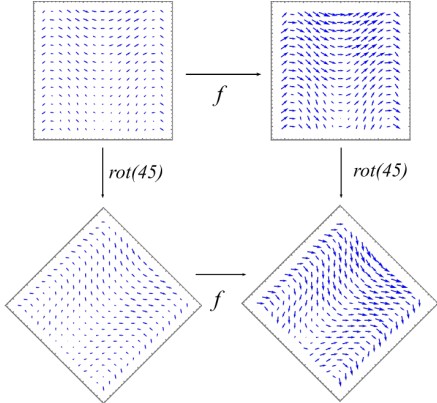

Specifically, for scale equivariance, we replace the convolution operation with group correlation over the group $G$ generated by translations *and* rescalings. Our method builds on that of Worrall and Welling [51], with significant novel adaptations to the physics domain: scaling affecting time, space, and magnitude; both up and down scaling; and scaling by any real number. For rotational symmetries, we leverage the key insight of Cohen and Welling [9] that the input, output, and hidden layers of the network are all acted upon by the symmetry group and thus should be treated as representations of the symmetry group. Our rotation-equivariant model is built using the flexible `E(2)-CNN` framework developed by Weiler and Cesa [49]. In the case of a uniform motion, or Galilean transformation, we show the above methods are too constrained. We use the simple but effective technique of convolutions conjugated by averaging operations.

Figure 1: Illustration of equivariance of e.g. $f(x) = 2x$ with respect to $T = \mathrm{rot}(\pi/4)$.

Research into equivariant neural networks has mostly been applied to tasks such as image classification and segmentation [27; 50; 49]. In contrast, we design equivariant networks in a completely different context, that of a time series representing a physical process. Forecasting high-dimensional turbulence is a significant step for equivariant neural networks compared to the low-dimensional physics examples and computer vision problems treated in other works.

We test on a simulated turbulent convection dataset and on real-world ocean current and temperature data. Ocean currents are difficult to predict using numerical methods due to unknown external forces and complex dynamics not fully captured by simplified mathematical models. These domains are chosen as examples, but since the symmetries we focus on are pervasive in almost all physics problems, we expect our techniques will be widely applicable. Our contributions include:

- We study the problem of improving the generalization capability and physical accuracy of deep learning models for learning complex physical dynamics such as turbulence and ocean currents.

- We design tailored methods with theoretical guarantees to incorporate various symmetries, including uniform motion, rotation, and scaling, into convolutional neural networks.

- When evaluated on turbulent convection and ocean current prediction, our models achieve significant improvement on generalization of both predictions and physical consistency.

- For different symmetries, our methods have an average 31% and maximum 78% reduction in energy error when evaluated on turbulent convection with no distributional shift.

## 2 MATHEMATICAL PRELIMINARIES

### 2.1 SYMMETRY GROUPS AND EQUIVARIANT FUNCTIONS

Formal discussion of symmetry relies on the concept of an abstract symmetry group. We give a brief overview, for a more formal treatment see Appendix A, or Lang [28].

A **group of symmetries** or simply **group** consists of a set $G$ together with a composition map $\circ\colon G \times G \to G$. The composition map is required to be associative and have an identity $1 \in G$. Most importantly, composition with any element of $G$ is required to be invertible.

Groups are abstract objects, but they become concrete when we let them act. A group $G$ has an **action** on a set $S$ if there is an action map $\cdot\colon G \times S \to S$ which is compatible with the composition law. We say further that $S$ is a $G$-**representation** if the set $S$ is a vector space and the group acts on $S$ by linear transformations.

**Definition 1** (invariant, equivariant). Let $f\colon X \to Y$ be a function and $G$ be a group. Assume $G$ acts on $X$ and $Y$. The function $f$ is $G$-**equivariant** if $f(gx) = gf(x)$ for all $x \in X$ and $g \in G$. The function $f$ is $G$-**invariant** if $f(gx) = f(x)$ for all $x \in X$ and $g \in G$.

## 2.2 PHYSICAL DYNAMICAL SYSTEMS

We investigate two dynamical systems: Rayleigh–Bénard convection and real-world ocean current and temperature. These systems are governed by Navier-Stokes equations.

**2D Navier-Stokes (NS) Equations.** Let $\boldsymbol{w}(\boldsymbol{x}, t)$ be the velocity vector field of a flow. The field $\boldsymbol{w}$ has two components $(u, v)$, velocities along the $x$ and $y$ directions. The governing equations for this physical system are the momentum equation, continuity equation, and temperature equation,

$$\frac{\partial \boldsymbol{w}}{\partial t} = -(\boldsymbol{w} \cdot \nabla)\boldsymbol{w} - \frac{1}{\rho_0}\nabla p + \nu\nabla^2 \boldsymbol{w} + f; \quad \nabla \cdot \boldsymbol{w} = 0; \quad \frac{\partial H}{\partial t} = \kappa\Delta H - (\boldsymbol{w} \cdot \nabla)H, \ (\mathcal{D}_{\mathrm{NS}})$$

where $H(\boldsymbol{x}, t)$ is temperature, $p$ is pressure, $\kappa$ is the heat conductivity, $\rho_0$ is initial density, $\alpha$ is the coefficient of thermal expansion, $\nu$ is the kinematic viscosity, and $f$ is the buoyant force.

## 2.3 SYMMETRIES OF DIFFERENTIAL EQUATIONS

By classifying the symmetries of a system of differential equations, the task of finding solutions is made far simpler, since the space of solutions will exhibit those same symmetries. Let $G$ be a group equipped with an action on 2-dimensional space $X = \mathbb{R}^2$ and 3-dimensional spacetime $\hat{X} = \mathbb{R}^3$. Let $V = \mathbb{R}^d$ be a $G$-representation. Denote the set of all $V$-**fields** on $\hat{X}$ as $\hat{\mathcal{F}}_V = \{\boldsymbol{w}\colon \hat{X} \to V : \boldsymbol{w} \text{ smooth}\}$. Define $\mathcal{F}_V$ similarly to be $V$-fields on $X$. Then $G$ has an induced action on $\hat{\mathcal{F}}_V$ by $(g\boldsymbol{w})(x, t) = g(\boldsymbol{w}(g^{-1}x, g^{-1}t))$ and on $\mathcal{F}_V$ analogously.

Consider a system of differential operators $\mathcal{D}$ acting on $\hat{\mathcal{F}}_V$. Denote the set of solutions $\mathrm{Sol}(\mathcal{D}) \subseteq \hat{\mathcal{F}}_V$. We say $G$ is **a symmetry group of** $\mathcal{D}$ if $G$ preserves $\mathrm{Sol}(\mathcal{D})$. That is, if $\varphi$ is a solution of $\mathcal{D}$, then for all $g \in G$, $g(\varphi)$ is also. In order to forecast the evolution of a system $\mathcal{D}$, we model the forward prediction function $f$. Let $\boldsymbol{w} \in \mathrm{Sol}(\mathcal{D})$. The input to $f$ is a collection of $k$ snapshots at times $t - k, \dots, t - 1$ denoted $\boldsymbol{w}_{t-i} \in \mathcal{F}_d$. The prediction function $f\colon \mathcal{F}_d^k \to \mathcal{F}_d$ is defined $f(\boldsymbol{w}_{t-k}, \dots, \boldsymbol{w}_{t-1}) = \boldsymbol{w}_t$. It predicts the solution at a time $t$ based on the solution in the past. Let $G$ be a symmetry group of $\mathcal{D}$. Then for $g \in G$, $g(\boldsymbol{w})$ is also a solution of $\mathcal{D}$. Thus $f(g\boldsymbol{w}_{t-k}, \dots, g\boldsymbol{w}_{t-1}) = g\boldsymbol{w}_t$. Consequently, $f$ is $G$-equivariant.

## 2.4 SYMMETRIES OF NAVIER-STOKES EQUATIONS

The Navier-Stokes equations are invariant under the following five different transformations. Individually each of these types of transformations generates a group of symmetries of the system. The full list of symmetry groups of NS equations and Heat equations are shown in Appendix B.6.

- Space translation:  $T_{\boldsymbol{c}}^{\mathrm{sp}}\boldsymbol{w}(\boldsymbol{x}, t) = \boldsymbol{w}(\boldsymbol{x} - \boldsymbol{c}, t), \ \ \boldsymbol{c} \in \mathbb{R}^2,$
- Time translation: $T_\tau^{\mathrm{time}}\boldsymbol{w}(\boldsymbol{x}, t) = \boldsymbol{w}(\boldsymbol{x}, t - \tau), \ \ \tau \in \mathbb{R},$
- Uniform motion: $T_{\boldsymbol{c}}^{\mathrm{um}}\boldsymbol{w}(\boldsymbol{x}, t) = \boldsymbol{w}(\boldsymbol{x}, t) + \boldsymbol{c}, \ \ \boldsymbol{c} \in \mathbb{R}^2,$
- Rotation/Reflection:  $T_R^{\mathrm{rot}}\boldsymbol{w}(\boldsymbol{x}, t) = R\boldsymbol{w}(R^{-1}\boldsymbol{x}, t), \ R \in O(2),$
- Scaling:  $T_\lambda^{sc}\boldsymbol{w}(\boldsymbol{x}, t) = \lambda\boldsymbol{w}(\lambda\boldsymbol{x}, \lambda^2 t), \ \ \lambda \in \mathbb{R}_{>0}.$

## 3 METHODOLOGY

We prescribe equivariance by training within function classes containing only equivariant functions. Our models can thus be theoretically guaranteed to be equivariant up to discretization error. We incorporate equivariance into two state-of-the-art architectures for dynamics prediction, `ResNet` and `U-net` [48]. Below, we describe how we modify the convolution operation in these models for different symmetries $G$ to form four $\mathtt{Equ}_G\mathtt{-ResNet}$ and four $\mathtt{Equ}_G\mathtt{-Unet}$ models.

### 3.1 EQUIVARIANT NETWORKS

The key to building equivariant networks is that the composition of equivariant functions is equivariant. Hence, if the maps between layers of a neural network are equivariant, then the whole network will be equivariant. Note that both the linear maps and activation functions must be equivariant. An important consequence of this principle is that the hidden layers must also carry a $G$-action. Thus, the hidden layers are not collections of scalar channels, but vector-valued $G$-representations.

**Equivariant Convolutions.** Consider a convolutional layer $\mathcal{F}_{\mathbb{R}^{d_{\text{in}}}} \to \mathcal{F}_{\mathbb{R}^{d_{\text{out}}}}$ with kernel $K$ from a $\mathbb{R}^{d_{\text{in}}}$-field to a $\mathbb{R}^{d_{\text{out}}}$-field. Let $\mathbb{R}^{d_{\text{in}}}$ and $\mathbb{R}^{d_{\text{out}}}$ be $G$-representations with action maps $\rho_{\text{in}}$ and $\rho_{\text{out}}$ respectively. Cohen et al. [11, Theorem 3.3] prove the network is $G$-equivariant if and only if

$$K(gv) = \rho_{\text{out}}^{-1}(g)K(v)\rho_{\text{in}}(g) \qquad \text{for all } g \in G. \tag{1}$$

A network composed of such equivariant convolutions is called a *steerable CNN*.

**Equivariant `ResNet` and `U-net`.** Equivariant `ResNet` architectures appear in [9; 10], and equivariant transposed convolution, a feature of `U-net`, is implemented in [49]. We prove in general that adding skip connections to a network does not affect its equivariance with respect to linear actions and also give a condition for `ResNet` or `Unet` to be equivariant in Appendix B.2.

**Relation to Data Augmentation.** To improve generalization, equivariant networks offer a better performing alternative to the popular technique of data augmentation [13]. Large symmetry groups normally require augmentation with many transformed examples. In contrast, for equivariant models, we have following proposition. (See Appendix B.1 for proof.)

**Proposition 1.** *$G$-equivariant models with equivariant loss learn equally (up to sample weight) from any transformation $g(s)$ of a sample $s$. Thus data augmentation does not help during training.*

### 3.2 TIME AND SPACE TRANSLATION EQUIVARIANCE

CNNs are time translation-equivariant as long as we predict in an autoregressive manner. Convolutional layers are also naturally space translation-equivariant (if cropping is ignored). Any activation function which acts identically pixel-by-pixel is equivariant.

### 3.3 ROTATIONAL EQUIVARIANCE

To incorporate rotational symmetry, we model $f$ using $\text{SO}(2)$-equivariant convolutions and activations within the `E(2)-CNN` framework of Weiler and Cesa [49]. In practice, we use the cyclic group $G = C_n$ instead of $G = \text{SO}(2)$ as for large enough $n$ the difference is practically indistinguishable due to space discretization. We use powers of the regular representation $\rho = \mathbb{R}[C_n]^m$ for hidden layers. The representation $\mathbb{R}[C_n]$ has basis given by elements of $C_n$ and $C_n$-action by permutation matrices. It has good descriptivity since it contains all irreducible representations of $C_n$, and it is compatible with any activation function applied channel-wise.

### 3.4 UNIFORM MOTION EQUIVARIANCE

Uniform motion is part of Galilean invariance and is relevant to all non-relativistic physics modeling. For a vector field $X \colon \mathbb{R}^2 \to \mathbb{R}^2$ and vector $c \in \mathbb{R}^2$, uniform motion transformation is adding a constant vector field to the vector field $X(v)$, $T_c^{\text{um}}(X)(v) = X(v) + c, c \in \mathbb{R}^2$. By the following corollary, proved in Appendix B.3, enforcing uniform motion equivariance as above by requiring all layers of the `CNN` to be equivariant severely limits the model.

**Corollary 2.** *If $f$ is a* CNN *alternating between convolutions $f_i$ and channel-wise activations $\sigma_i$ and the combined layers $\sigma_i \circ f_i$ are uniform motion equivariant, then $f$ is affine.*

To overcome this limitation, we relax the requirement by conjugating the model with shifted input distribution. For each sliding local block in each convolutional layer, we shift the mean of input tensor to zero and shift the output back after convolution and activation function per sample. In other words, if the input is $\mathcal{P}_{b \times d_{in} \times s \times s}$ and the output is $\mathcal{Q}_{b \times d_{out}} = \sigma(\mathcal{P} \cdot K)$ for one sliding local block, where $b$ is batch size, $d$ is number of channels, $s$ is the kernel size, and $K$ is the kernel, then

$$\boldsymbol{\mu}_i = \text{Mean}_{jkl}\left(\mathcal{P}_{ijkl}\right); \quad \mathcal{P}_{ijkl} \mapsto \mathcal{P}_{ijkl} - \boldsymbol{\mu}_i; \quad \mathcal{Q}_{ij} \mapsto \mathcal{Q}_{ij} + \boldsymbol{\mu}_i. \tag{2}$$

This will allow the convolution layer to be equivariant with respect to uniform motion. If the input is a vector field, we apply this operation to each element.

**Proposition 3.** *A residual block $f(\boldsymbol{x}) + \boldsymbol{x}$ is uniform motion equivariant if the residual connection $f$ is uniform motion invariant.*

By the proposition 3 above that is proved in Appendix B.3, within ResNet, residual mappings should be *invariant*, not equivariant, to uniform motion. That is, the skip connection $f^{(i,i+2)} = I$ is equivariant and the residual function $f^{(i,i+1)}$ should be invariant. Hence, for the first layer in each residual block, we omit adding the mean back to the output $\mathcal{Q}_{ij}$. In the case of Unet, when upscaling, we pad with the mean to preserve the overall mean.

### 3.5 Scale Equivariance

Scale equivariance in dynamics is unique as the physical law dictates the scaling of magnitude, space and time simultaneously. This is very different from scaling in images regarding resolutions [51]. For example, the Navier-Stokes equations are preserved under a specific scaling ratio of time, space, and velocity given by the transformation

$$T_\lambda \colon \boldsymbol{w}(\boldsymbol{x}, t) \mapsto \lambda \boldsymbol{w}(\lambda \boldsymbol{x}, \lambda^2 t), \tag{3}$$

where $\lambda \in \mathbb{R}_{>0}$. We implement two different approaches for scale equivariance, depending on whether we tie the physical scale with the resolution of the data.

**Resolution Independent Scaling.** We fix the resolution and scale the magnitude of the input by varying the discretization step size. An input $\boldsymbol{w} \in \mathcal{F}_{\mathbb{R}^2}^k$ with step size $\Delta_x(\boldsymbol{w})$ and $\Delta_t(\boldsymbol{w})$ can be scaled $\boldsymbol{w}' = T_\lambda^{sc}(\boldsymbol{w}) = \lambda \boldsymbol{w}$ by scaling the magnitude of vector alone, provided the discretization constants are now assumed to be $\Delta_x(\boldsymbol{w}') = 1/\lambda \Delta_x(\boldsymbol{w})$ and $\Delta_t(\boldsymbol{w}') = 1/\lambda^2 \Delta_t(\boldsymbol{w})$. We refer to this as *magnitude* equvariance hereafter.

To obtain magnitude equivariance, we divide the input tensor by the MinMax scaler (the maximum of the tensor minus the minimum) and scale the output back after convolution and activation per sliding block. We found that the standard deviation and mean L2 norm may work as well but are not as stable as the MinMax scaler. Specifically, using the same notation as in Section 3.4,

$$\boldsymbol{\sigma}_i = \text{MinMax}_{jkl}\left(\mathcal{P}_{ijkl}\right); \quad \mathcal{P}_{ijkl} \mapsto \mathcal{P}_{ijkl}/\boldsymbol{\sigma}_i; \quad \mathcal{Q}_{ij} \mapsto \mathcal{Q}_{ij} \cdot \boldsymbol{\sigma}_i. \tag{4}$$

**Resolution Dependent Scaling.** If the physical scale of the data is fixed, then scaling corresponds to a change in resolution and time step size. To achieve this, we replace the convolution layers with group correlation layers over the group $G = (\mathbb{R}_{>0}, \cdot) \ltimes (\mathbb{R}^2, +)$ of scaling and translations. In convolution, we translate a kernel $K$ across an input $\boldsymbol{w}$ as such $\boldsymbol{v}(\boldsymbol{p}) = \sum_{\boldsymbol{q} \in \mathbb{Z}^2} \boldsymbol{w}(\boldsymbol{p} + \boldsymbol{q})K(\boldsymbol{q})$. The $G$-correlation upgrades this operation by both translating *and* scaling the kernel relative to the input,

$$\boldsymbol{v}(\boldsymbol{p}, s, \mu) = \sum_{\lambda \in \mathbb{R}_{>0}, t \in \mathbb{R}, \boldsymbol{q} \in \mathbb{Z}^2} \lambda \boldsymbol{w}(\lambda \boldsymbol{p} + \boldsymbol{q}, \lambda^2 t, \lambda \mu)K(\boldsymbol{q}, s, t, \lambda), \tag{5}$$

where $s$ and $t$ denote the indices of output and input channels respectively. We add an axis to the tensors corresponding the scale factor $\mu$. Note that we treat the channel as a time dimension both with respective to our input and scaling action. As a consequence, as the number of channels increases in the lower layers of Unet and ResNet, the temporal resolution increases, which is analogous to temporal refinement in numerical methods [24; 31]. For the input $\tilde{\boldsymbol{w}}$ of first layer where $\tilde{\boldsymbol{w}}$ has no levels originally, $\boldsymbol{w}(p, s, \lambda) = \lambda \tilde{\boldsymbol{w}}(\lambda p, \lambda^2 s)$.

Our model builds on the methods of Worrall and Welling [51], but with important adaptations for the physical domain. Our implementation of group correlation equation 5 directly incorporates the physical scaling law equation 3 of the system equation $\mathcal{D}_{NS}$. This affects time, space, and magnitude. (For heat, we drop the magnitude scaling.) The physical scaling law dictates our model should be equivariant to both up and down scaling and by any $\lambda \in \mathbb{R}_{>0}$. Practically, the sum is truncated to 7 different $1/3 \leq \lambda \leq 3$ and discrete data is continuously indexed using interpolation. Note equation 3 demands we scale *anisotropically*, i.e. differently across time and space.

## 4  RELATED WORK

**Equivariance and Invariance.**  Developing neural nets that preserve symmetries has been a fundamental task in image recognition [12; 49; 9; 7; 29; 27; 3; 52; 10; 19; 50; 16; 42]. But these models have never been applied to forecasting physical dynamics. Jaiswal et al. [23]; Moyer et al. [37] proposed approaches to find representations of data that are invariant to changes in specified factors, which is different from our physical symmetries. Ling et al. [30] and Fang et al. [17] studied tensor invariant neural networks to learn the Reynolds stress tensor while preserving Galilean invariance, and Mattheakis et al. [34] embedded even/odd symmetry of a function and energy conservation into neural networks to solve differential equations. But these two papers are limited to fully connected neural networks. Sosnovik et al. [44] extend Worrall and Welling [51] to group correlation convolution. But these two papers are limited to 2D images and are not magnitude equivariant, which is still inadequate for fluid dynamics. Bekkers [4] describes principles for endowing a neural architecture with invariance with respect to a Lie group.

**Physics-informed Deep Learning.**  Deep learning models have been used often to model physical dynamics. For example, Wang et al. [48] unified the CFD technique and U-net to generate predictions with higher accuracy and better physical consistency. Kim and Lee [25] studied unsupervised generative modeling of turbulent flows but the model is not able to make real time future predictions given the historic data. Anderson et al. [1] designed rotationally covariant neural network for learning molecular systems. Raissi et al. [40; 41] applied deep neural networks to solve PDEs automatically but these approaches require explicit input of boundary conditions during inference, which are generally not available in real-time. Mohan et al. [35] proposed a purely data-driven DL model for turbulence, but the model lacks physical constraints and interpretability. Wu et al. [53] and Beucler et al. [5] introduced statistical and physical constraints in the loss function to regularize the predictions of the model. However, their studies only focused on spatial modeling without temporal dynamics. Morton et al. [36] incorporated Koopman theory into a encoder-decoder architecture but did not study the symmetry of fluid dynamics.

**Video Prediction.**  Our work is related to future video prediction. Conditioning on the observed frames, video prediction models are trained to predict future frames, e.g., [33; 18; 54; 47; 39; 18]. Many of these models are trained on natural videos with complex noisy data from unknown physical processes. Therefore, it is difficult to explicitly incorporate physical principles into these models. Our work is substantially different because we do not attempt to predict object or camera motions.

## 5  EXPERIMENTS

We test our models on Rayleigh-Bénard convection and real-world ocean currents. We also evaluated on the heat diffusion systems, see Appendix C for more results. The implementation details and a detailed description of energy spectrum error can be found in Appendices D and B.7.

**Evaluation Metrics.**  Our goal is to show that adding symmetry improves both the accuracy and the physical consistency of predictions. For accuracy, we use Root Mean Square Error (RMSE) between the forward predictions and the ground truth over all pixels. For physical consistency, we calculate the Energy Spectrum Error (ESE) which is the RMSE of the log of energy spectrum. ESE can indicate whether the predictions preserve the correct statistical distributions of the fluids and obey the energy conservation law, which is a critical metric for physical consistency.

**Experimental Setup.**  `ResNet`[20] and `U-net`[43] are the best-performing models for our tasks [48] and are well-suited for our tasks. Thus, we implemented these two convolutional architectures equipped with four different symmetries, which we name `Equ-ResNet(U-net)`. We use a rolling window approach to generate sequences with step size 1 for the RBC data and step size 3 for the

Table 2: The RMSE and ESE of the `ResNet(Unet)` and four `Equ-ResNets(Unets)` predictions on the original and four transformed test sets of Rayleigh-Bénard Convection. `Augm` is `ResNet(Unet)` trained on the augmented training set with additional samples applied with random transformations from the relevant symmetry group. Each column contains all models' prediction errors on the original test set and four different transformed test sets.

| | Root Mean Square Error($10^3$) | | | | | Energy Spectrum Errors | | | | |
|---|---|---|---|---|---|---|---|---|---|---|
| | *Orig* | *UM* | *Mag* | *Rot* | *Scale* | *Orig* | *UM* | *Mag* | *Rot* | *Scale* |
| **ResNet** | 0.67±0.24 | 2.94±0.84 | 4.30±1.27 | 3.46±0.39 | 1.96±0.16 | 0.46±0.19 | 0.56±0.29 | 0.26±0.14 | 1.59±0.42 | 4.32±2.33 |
| **Augm** | | 1.10±0.20 | 1.54±0.12 | 0.92±0.09 | 1.01±0.11 | | 1.37±0.02 | 1.14±0.32 | 1.92±0.21 | 1.55±0.14 |
| Equ$_\text{UM}$ | 0.71±0.26 | **0.71±0.26** | | | | 0.33±0.11 | **0.33±0.11** | | | |
| Equ$_\text{Mag}$ | 0.69±0.24 | | **0.67±0.14** | | | 0.34±0.09 | | **0.19±0.02** | | |
| Equ$_\text{Rot}$ | **0.65±0.26** | | | **0.76±0.02** | | 0.31±0.06 | | | **1.23±0.04** | |
| Equ$_\text{Scal}$ | 0.70±0.02 | | | | **0.85±0.09** | 0.44±0.22 | | | | **0.68±0.26** |
| **U-net** | 0.64±0.24 | 2.27±0.82 | 3.59±1.04 | 2.78±0.83 | 1.65±0.17 | 0.50±0.04 | 0.34±0.10 | 0.55±0.05 | 0.91±0.27 | 4.25±0.57 |
| **Augm** | | 0.75±0.28 | 1.33±0.33 | 0.86±0.04 | 1.11±0.07 | | 0.96±0.23 | 0.44±0.21 | 1.24±0.04 | 1.47±0.11 |
| Equ$_\text{UM}$ | 0.68±0.26 | **0.71±0.24** | | | | 0.23±0.06 | **0.14±0.05** | | | |
| Equ$_\text{Mag}$ | 0.67±0.11 | | **0.68±0.14** | | | 0.42±0.04 | | **0.34±0.06** | | |
| Equ$_\text{Rot}$ | 0.68±0.25 | | | **0.74±0.01** | | **0.11±0.02** | | | **1.16±0.05** | |
| Equ$_\text{Scal}$ | 0.69±0.13 | | | | **0.90±0.25** | 0.45±0.32 | | | | **0.89±0.29** |

ocean data. All models predict raw velocity and temperature fields up to 10 steps ahead autoregressively. We use the MSE loss function that accumulates the forecasting errors. We split the data 60%-20%-20% for training-validation-test across time and report mean errors over five random runs.

## 5.1 EQUIVARIANCE ERRORS

The equivariance errors can be defined as $\text{EE}_T(x) = |T(f(x)) - f(T(x))|$, where $x$ is an input, $f$ is a neural net, $T$ is a transformation from a symmetry group. We empirically measure the equivariance errors of all equivariant models we have designed. Table 1 shows the equivariance errors of `ResNet` and `Equ-ResNet`. The transformation $T$ is sampled in the same way as we generated the transformed Rayleigh-Bénard Convection test sets. See more details in Appendix B.5.

## 5.2 EXPERIMENTS ON SIMULATED RAYLEIGH-BÉNARD CONVECTION DYNAMICS

**Data Description.** Rayleigh-Bénard Convection occurs in a horizontal layer of fluid heated from below and is a major feature of the El Niño dynamics. The dataset comes from two-dimensional turbulent flow simulated using the Lattice Boltzmann Method [8] with Rayleigh number $2.5 \times 10^8$. We divide each $1792 \times 256$ image into 7 square subregions of size $256 \times 256$, then downsample to $64 \times 64$ pixels. To test the models' generalization ability, we generate additional four test sets : 1) *UM*: added random vectors drawn from $U(-1, 1)$; 2) *Mag*: multiplied by random values sampled from $U(0, 2)$; 3) *Rot*: randomly rotated by the multiples of $\pi/2$; 4) *Scale*: scaled by $\lambda$ sampled from $U(1/5, 2)$. Due to lack of a fixed reference frame, real-world data would be transformed relative to training data. We use transformed data to mimic this scenario.

**Prediction Performance.** Table 2 shows the prediction RMSE and ESE on the original and four transformed test sets by the non-equivariant `ResNet(Unet)` and four `Equ-ResNets(Unets)`. `Augm` is `ResNet(Unet)` trained on the augmented training set with additional samples with random transformations applied from the relevant symmetry group. The augmented training set contains additional transformed samples and is three times the size of the original training set. Each column contains the prediction errors by the non-equivariant and equivariant models on each test set. On the original test set, all models have similar RMSE, yet

Table 1: Equivariance Errors of `ResNet(Unets)` and `Equ-ResNet(Unets)`.

| $\text{EE}_T(10^3)$ | *UM* | *Mag* | *Rot* | *Scale* |
|---|---|---|---|---|
| `ResNets` | 2.010 | 1.885 | 5.895 | 1.658 |
| Equ$_\text{ResNets}$ | 0.0 | 0.0 | 1.190 | 0.579 |
| `Unets` | 1.070 | 0.200 | 1.548 | 1.809 |
| Equ$_\text{Unets}$ | 0.0 | 0.0 | 0.794 | 0.481 |

the equivariant models have lower ESE. This demonstrates that incorporating symmetries preserves the representation powers of CNNs and even improves models' physical consistency.

On the transformed test sets, we can see that `ResNet(Unet)` fails, while `Equ-ResNets(Unets)` performs even much better than `Augm-ResNets(Unets)`. This demonstrates the value of equivariant models over data augmentation for improving generalization. Figure 2 shows the ground truth

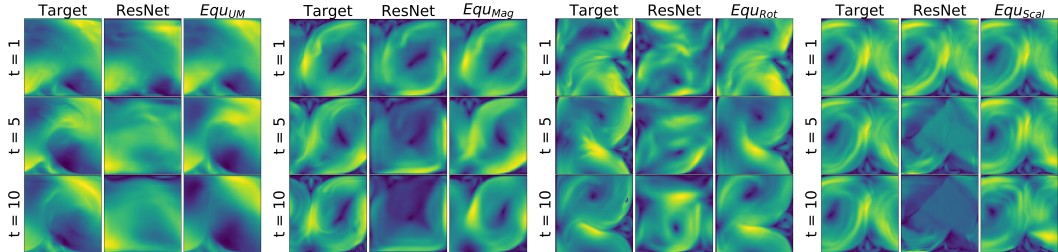

Figure 2: The ground truth and the predicted velocity norm fields $\|\boldsymbol{w}\|_2$ at time step 1, 5 and 10 by the `ResNet` and four `Equ-ResNets` on the four transformed test samples. The first column is the target, the second is `ResNet` predictions, and the third is predictions by `Equ-ResNets`.

and the predicted velocity fields at time step 1, 5 and 10 by the `ResNet` and four `Equ-ResNets` on the four transformed test samples.

**Generalization.** In order to evaluate models' generalization ability with respect to the extent of distributional shift, we created additional test sets with different scale factors from $\frac{1}{5}$ to 1. Figure 3 shows `ResNet` and $\text{Equ}_{\text{Scal}}$-`ResNet` prediction RMSEs (left) and ESEs (right) on the test sets upscaled by different factors. We observed that $\text{Equ}_{\text{Scal}}$-`ResNet` is very robust across various scaling factors while `ResNet` does not generalize.

We also compare `ResNet` and `Equ-ResNet` when both train and test sets have random transformations from the relevant symmetry group applied to each sample. This mimics real-world data in which each sample has unknown reference frame. As shown in Table 3 shows `Equ-ResNet` outperforms `ResNet` on average by 34% RMSE and 40% ESE.

Table 3: Performance comparison on transformed train and test sets.

|  | RMSE | ESE |
|---|---|---|
| **ResNet** | 1.03±0.05 | 0.96±0.10 |
| $\text{Equ}_{\text{UM}}$ | **0.69±0.01** | **0.35±0.13** |
| **ResNet** | 1.50±0.02 | 0.55±0.11 |
| $\text{Equ}_{\text{Mag}}$ | **0.75±0.04** | **0.39±0.02** |
| **ResNet** | 1.18±0.05 | 1.21±0.04 |
| $\text{Equ}_{\text{Rot}}$ | **0.77±0.01** | **0.68±0.01** |
| **ResNet** | 0.92±0.01 | 1.34±0.07 |
| $\text{Equ}_{\text{Scal}}$ | **0.74±0.03** | **1.02±0.02** |

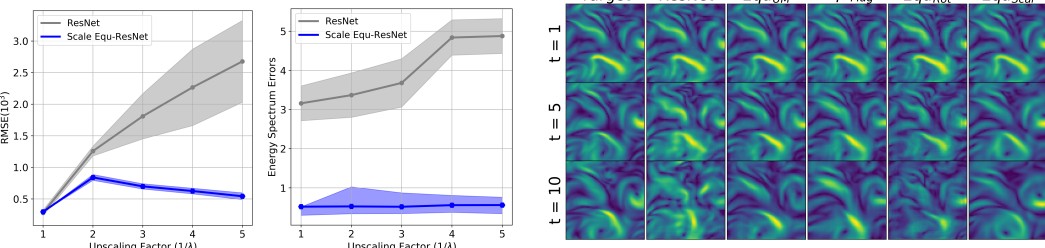

Figure 3: Left: Prediction RMSE and ESE over five runs of `ResNet` and $\text{Equ}_{\text{Scal}}$-`ResNet` on the Rayleigh-Bénard Convection test set upscaled by different factors. Right: The ground truth and predicted ocean currents $\|\boldsymbol{w}\|_2$ by `ResNet` and four `Equ-ResNets` on the test set of future time.

## 5.3 EXPERIMENTS ON REAL WORLD OCEAN DYNAMICS

**Data Description.** We use the reanalysis ocean current velocity data generated by the NEMO ocean engine [32].[1] We selected an area from each of the Atlantic, Indian and North Pacific Oceans from 01/01/2016 to 08/18/2017 and extracted 64×64 sub-regions for our experiments. The corresponding latitude and longitude ranges for the selected regions are (-44~-23, 25~46), (55~76, -39~-18) and (-174~-153, 5~26) respectively. We not only test all models on the future data but also on a different domain (-180~-159, -40~-59) in South Pacific Ocean from 01/01/2016 to 12/15/2016.

**Prediction Performance.** Table 4 shows the RMSE and ESE of `ResNets(Unets)`, and equivariant `Equ-ResNets(Unets)` on the test sets with different time range and spatial domain from

---

[1]The data are available at `https://resources.marine.copernicus.eu/?option=com_csw&view=details&product_id=GLOBAL_ANALYSIS_FORECAST_PHY_001_024`

the training set. All the equivariant models outperform the non-equivariant baseline on RMSE, and `Equ_Scal`-ResNet achieves the lowest RMSE. For ESE, only the `Equ_Mag`-ResNet(Unet) is worse than the baseline. Also, it is remarkable that the `Equ_Rot` models have significantly lower ESE than others, suggesting that they correctly learn the statistical distribution of ocean currents.

**Comparison with Data Augmentation.** We also compare `Equ-ResNets(Unets)` `ResNets(Unets)` that are trained with data-augmentation (`Augm`) in Table 4. In all cases, equivariant models outperforms the baselines trained with data augmentation. We find that data augmentation sometimes improves slightly on RMSE but not as much as the equivariant models. And, in fact, ESE is uniformly worse for models trained with data augmentation than even the baselines. In contrast, the equivariant models have much better ESE than the baselines with or without augmentation. We believe data augmentation presents a trade-off in learning. Though the model may be less sensitive to the various transformations we consider, we need to train bigger models longer on many more samples. The models may not have enough capacity to learn the symmetry from the augmented data and the dynamics of the fluids at the same time. By comparison, equivariant architectures do not have this issue.

Figure 3 shows the ground truth and the predicted ocean currents at time step $1, 5, 10$ by different models. We can see that equivariant models' predictions are more accurate and contain more details than the baselines. Thus, incorporating symmetry into deep learning models can improve the prediction accuracy of ocean currents. The most recent work on this dataset is de Bezenac et al. [15], which combines a warping scheme and a `U-net` to predict temperature. Since our models can also be applied to advection-diffusion systems, we also investigated the task of ocean temperature field predictions. We observe that `Equ_UM`-Unet performs slightly better than de Bezenac et al. [15]. For additional results, see Appendix E.

Table 4: Prediction RMSE and ESE comparison on the two ocean currents test sets.

| | RMSE | | ESE | |
|---|---|---|---|---|
| | $Test_{time}$ | $Test_{domain}$ | $Test_{time}$ | $Test_{domain}$ |
| **ResNet** | 0.71±0.07 | 0.72±0.04 | 0.83±0.06 | 0.75±0.11 |
| Augm_UM | 0.70±0.01 | 0.70±0.07 | 1.06±0.06 | 1.06±0.04 |
| Augm_Mag | 0.76±0.02 | 0.71±0.01 | 1.08±0.08 | 1.05±0.8 |
| Augm_Rot | 0.73±0.01 | 0.69±0.01 | 0.94±0.01 | 0.86±0.01 |
| Augm_Scal | 0.97±0.06 | 0.92±0.04 | 0.85±0.03 | 0.95±0.11 |
| Equ_UM | 0.68±0.06 | 0.68±0.16 | 0.75±0.06 | 0.73±0.08 |
| Equ_Mag | 0.66±0.14 | **0.68±0.11** | 0.84±0.04 | 0.85±0.14 |
| Equ_Rot | 0.69±0.01 | 0.70±0.08 | **0.43±0.15** | **0.28±0.20** |
| Equ_Scal | **0.63±0.02** | 0.68±0.21 | 0.44±0.05 | 0.42±0.12 |
| **U-net** | 0.70±0.13 | 0.73±0.10 | 0.77±0.12 | 0.73±0.07 |
| Augm_UM | 0.68±0.02 | 0.68±0.01 | 0.85±0.04 | 0.83±0.04 |
| Augm_Mag | 0.69±0.02 | 0.67±0.10 | 0.78±0.03 | 0.86±0.02 |
| Augm_Rot | 0.79±0.01 | 0.70±0.01 | 0.79±0.01 | 0.78±0.02 |
| Augm_Scal | 0.71±0.01 | 0.77±0.02 | 0.84±0.01 | 0.77±0.02 |
| Equ_UM | 0.66±0.10 | 0.67±0.03 | 0.73±0.03 | 0.82±0.13 |
| Equ_Mag | **0.63±0.08** | **0.66±0.09** | 0.74±0.05 | 0.79±0.04 |
| Equ_Rot | 0.68±0.05 | 0.69±0.02 | **0.42±0.02** | 0.47±0.07 |
| Equ_Scal | 0.65±0.09 | 0.69±0.05 | 0.45±0.13 | **0.43±0.05** |

## 6 CONCLUSION AND FUTURE WORK

We develop methods to improve the generalization of deep sequence models for learning physical dynamics. We incorporate various symmetries by designing equivariant neural networks and demonstrate their superior performance on 2D time series prediction both theoretically and experimentally. Our designs obtain improved physical consistency for predictions. In the case of transformed test data, our models generalize significantly better than their non-equivariant counterparts. Importantly, all of our equivariant models can be combined and can be extended to 3D cases. The group $G$ also acts on the boundary conditions and external forces of a system $\mathcal{D}$. If these are $G$-invariant, then the system $\mathcal{D}$ is strictly invariant as in Section 2.3. If not, one must consider a family of solutions $\cup_{g \in G} \text{Sol}(g\mathcal{D})$ to retain equivariance. To the best of our best knowledge, there does not exist a single model with equivariance to the full symmetry group of the Navier-Stokes equations. It is possible but non-trivial, and we continue to work on combining different equivariances. Future work also includes speeding up the the scale-equivariant models and incorporating other symmetries into DL models.

### ACKNOWLEDGMENTS

This work was supported in part by Google Faculty Research Award, NSF Grant #2037745, and the U. S. Army Research Office under Grant W911NF-20-1-0334. The Titan Xp used for this research was donated by the NVIDIA Corporation. This research used resources of the National Energy Research Scientific Computing Center, a DOE Office of Science User Facility supported by the Office of Science of the U.S. Department of Energy under Contract No. DE-AC02-05CH11231. We also thank Dragos Bogdan Chirila for providing the turbulent flow data.

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

## A    ADDITIONAL BACKGROUND ON GROUP THEORY

We give a brief overview of group theory and representation theory. For a more complete introduction to the topic see Lang [28]. We start with the definition of an abstract symmetry group.

**Definition 2** (group). A group of symmetries or simply *group* is a set $G$ together with a binary operation $\circ\colon G \times G \to G$ called *composition* satisfying three properties:

1. (*identity*) There is an element $1 \in G$ such that $1 \circ g = g \circ 1 = g$ for all $g \in G$,

2. (*associativity*) $(g_1 \circ g_2) \circ g_3 = g_1 \circ (g_2 \circ g_3)$ for all $g_1, g_2, g_3 \in G$,

3. (*inverses*) if $g \in G$, then there is an element $g^{-1} \in G$ such that $g \circ g^{-1} = g^{-1} \circ g = 1$.

**Definition 3** (Lie group). A group $G$ is a *Lie group* if it is also a smooth manifold over $\mathbb{R}$ and the composition and inversion maps are *smooth*, i.e. infinitely differentiable.

**Example 1.** Let $G = GL_2(\mathbb{R})$ be the set of $2 \times 2$ invertible real matrices. The set is closed under inversion and matrix multiplication gives a well-defined composition. This a 4-dimensional real Lie group.

**Example 2.** Let $G = D_3 = \{1, r, r^2, s, rs, r^2s\}$ where $r$ is rotation by $2\pi/3$ and $s$ is reflection over the $y$-axis. This is the group of symmetries of an equilateral triangle pointing along the $y$-axis, see Figure 2.

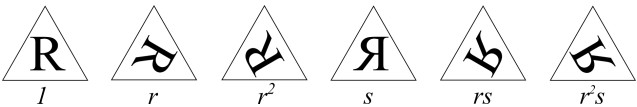

Figure 4: Illustration of $D_3$ acting on a triangle with the letter "R".

Groups are abstract objects, but they become concrete when we let them act.

**Definition 4** (action). A group $G$ acts on a set $S$ if there is an action map $\cdot\colon G \times S \to S$ satisfying

1. $1 \cdot x = x$ for all $x \in S$, $g \in G$,

2. $g_1 \cdot (g_2 \cdot x) = (g_1 \circ g_2) \cdot x$ for all $x \in S$, $g_1, g_2 \in G$.

**Definition 5** (representation). We say $S$ is a *$G$-representation* if $S$ is an $\mathbb{R}$-vector space and $G$ acts on $S$ by linear transformations, that is,

1. $g \cdot (x + y) = g \cdot x + g \cdot y$ for all $x, y \in S$, $g \in G$,

2. $g \cdot (cx) = c(g \cdot x)$ for all $x \in S$, $g \in G$, $c \in \mathbb{R}$.

**Example 3.** The group $D_3$ acts on $S$, the set of points in an equilateral triangle, as in Figure 2. The vector space $\mathbb{R}^2$ is both a $D_3$-representation and a $GL_2(\mathbb{R})$-representation.

The language of group theory allows us to formally define equivariance and invariance.

**Definition 6** (invariant, equivariant). Let $f\colon X \to Y$ be a function and $G$ be a group.

1. Assume $G$ acts on $X$. The function $f$ is *$G$-invariant* if $f(gx) = x$ for all $x \in X$ and $g \in G$.

2. Assume $G$ acts on $X$ and $Y$. The function $f$ is *$G$-equivariant* if $f(gx) = gf(x)$ for all $x \in X$ and $g \in G$.

See Figure 1 for an illustration. Note that we often omit the different action maps of $G$ on $X$ and on $Y$ in our notion when they are clear from context.

We can combine and decompose representations in different ways.

**Definition 7** (direct sum, tensor product). Let $V$ and $W$ be $G$-representations.

1. The *direct sum* $V \oplus W$ has underlying set $V \times W$. As a vector space it has scalars $c(v, w) = (cv, cw)$ and addition $(v_1, w_1) + (v_2, w_2) = (v_1 + v_2, w_1 + w_2)$. It is a $G$-representation with action $g \cdot (v, w) = (gv, gw)$.

2. The *tensor product*

$$V \otimes W = \left\{ \sum_i v_i \otimes w_i : v_i \in V, w_i \in W \right\}$$

is a $G$-representation with action $g \cdot v \otimes w = (gv) \otimes (gw)$.

**Definition 8** (irreducible). Let $V$ be a $G$-representation.

1. If $W$ is a subspace of $V$ and is closed under the action of $G$, i.e. $gw \in W$ for all $w \in W, g \in G$, then we say it is a *subrepresentation*.

2. If $0$ and $V$ itself are the only subrepresentations of $V$, then it is *irreducible*.

Irreducible representations are the "prime" building blocks of representations. A **compact** Lie group is one which is closed and bounded. The rotation group $SO(2, \mathbb{R})$ is compact, but the group $(\mathbb{R}, +)$ is not. All finite groups are also compact Lie groups. The following theorem vastly simplifies our understanding of possible representations of compact Lie groups (see e.g. Knapp [26]).

**Theorem 4** (Weyl's Complete Reducibility Theorem). *Let $G$ be a compact real Lie group. Every finite-dimensional representation of $V$ is a direct sum of irreducible representations $V = \oplus_i V_i$.*

Thus to classify the possible finite-dimensional representations of $G$, one need only to find all possible irreducible representations of $G$.

# B ADDITIONAL THEORY

## B.1 EQUIVARIANT NETWORKS AND DATA AUGMENTATION

A classic strategy for dealing with distributional shift by transformations in a group $G$ is to augment the training set $\mathcal{S}$ by adding samples transformed under $G$. That is, using the new training set $\mathcal{S}' = \bigcup_{g \in G} g(S)$. We show that data augmentation has no advantage for a perfectly equivariant parameterized function $f_\theta(x)$ since training samples $(x, y)$ and $(gx, gy)$ are equivalent. That is, $f_\theta$ learns the same from $(x, y)$ as from $(gx, gy)$ but with only possibly different sample weight. The following is a more formal statement of Proposition 1.

**Proposition 5.** *Let $G$ act on $X$ and $Y$. Let $f_\theta \colon X \to Y$ be a parameterized class of $G$-equivariant functions differentiable with respect to $\theta$. Let $\mathcal{L} \colon Y \times Y \to \mathbb{R}$ be a $G$-equivariant loss function where $G$ acts on $\mathbb{R}$ by $\chi$, we have,*

$$\chi(g) \nabla_\theta \mathcal{L}(f_\theta(x), y) = \nabla_\theta \mathcal{L}(f_\theta(gx), gy).$$

*Proof.* Equality of the gradients follows equality of the functions $\mathcal{L}(f_\theta(gx), gy) = \chi(g)\mathcal{L}(g^{-1}f_\theta(gx), y) = \chi(g)\mathcal{L}(f_\theta(x), y)$. $\square$

In the case of RMSE and rotation or uniform motion, the loss function is invariant. That is, equivariant with $\chi(g) = 1$. Thus the gradient for sample $(x, y)$ and $(gx, gy)$ is equal. In the case of scale, the loss function is equivariant with $G = (\mathbb{R}_{>0}, \cdot)$ and $\chi(\lambda) = \lambda$. In that case, the sample $(gx, gy)$ is the same as the sample $(x, y)$ but with sample weight $\chi(g)$.

## B.2 ADDING SKIP CONNECTIONS PRESERVES EQUIVARIANCE

We prove in general that adding skip connections to a network does not affect its equivariance with respect to linear actions in the following proposition 6. Define $f^{(ij)}$ as the functional mapping between layer $i$ and layer $j$.

**Proposition 6.** *Let the layer $V^{(i)}$ be a $G$-representations for $0 \le i \le n$. Let $f^{(ij)} \colon V^{(i)} \to V^{(j)}$ be $G$-equivariant for $i < j$. Define recursively $\boldsymbol{x}^{(j)} = \sum_{0 \le i < j} f^{(ij)}(\boldsymbol{x}^{(i)})$. Then $\boldsymbol{x}^{(n)} = f(\boldsymbol{x}^{(0)})$ is $G$-equivariant.*

*Proof.* Assume $\boldsymbol{x}^{(i)}$ is an equivariant function of $\boldsymbol{x}^{(0)}$ for $i < j$. Then by equivariance of $f^{(ij)}$ and by linearity of the $G$-action,

$$\sum_{0 \leq i < j} f^{(ij)}(g\boldsymbol{x}^{(i)}) = \sum_{0 \leq i < j} gf^{(ij)}(\boldsymbol{x}^{(i)}) = g\boldsymbol{x}^{(j)},$$

for $g \in G$. By induction, $\boldsymbol{x}^{(n)} = f(\boldsymbol{x}^{(0)})$ is equivariant with respect to $G$. $\qquad\square$

Both `ResNet` and `U-net` may be modeled as in Proposition 6 with some convolutional and activation components $f^{(i,i+1)}$ and some skip connections $f^{(ij)} = I$ with $j - i \geq 2$. Since $I$ is equivariant for any $G$, we thus have:

**Corollary 7.** *If the layers of* `ResNet` *or* `U-net` *are $G$-representations and the convolutional mappings and activation functions are $G$-equivariant, then the entire network is $G$-equivariant.* $\quad\square$

Corollary 7 allows us to build equivariant convolutional networks for rotational and scaling transformations, which are linear actions.

### B.3  RESULTS ON UNIFORM MOTION EQUIVARIANCE

In this section, we prove that for the combined convolution-activation layers of a CNN to be uniform motion equivariant, the CNN must be an affine function. We assume that the activation function is applied pointwise. That is, the same activation function is applied to every one-dimensional channel independently.

**Proposition 8.** *Let $X$ be a tensor of shape $h \times w \times c$ and $K$ be convolutional kernel of shape $k \times k \times c$. Let $f(\boldsymbol{X}) = \boldsymbol{X} * K$ be a convolutional layer which is equivariant with respect to arbitrary uniform motion $\boldsymbol{X} \mapsto \boldsymbol{X} + \boldsymbol{C}$ for $\boldsymbol{C}$ a constant tensor of the same shape as $\boldsymbol{X}$. That is $C_{ijk} = c$ for all $i, j, k$ for some fixed $c \in \mathbb{R}$. Then the sum of the weights of $K$ is 1.*

*Proof.* Since $f$ is equivariant, $\boldsymbol{X} * K + \boldsymbol{C} = (\boldsymbol{X} + \boldsymbol{C}) * K$. By linearity, $\boldsymbol{C} * K = \boldsymbol{C}$. Then because $\boldsymbol{C}$ is a constant vector field, $\boldsymbol{C} * K = \boldsymbol{C}(\sum_v K(v))$. As $\boldsymbol{C}$ is arbitrary, $\sum_v K(v) = 1$. $\qquad\square$

For an activation function to be uniform motion equivariant, it must be a translation.

**Proposition 9.** *Let $\sigma \colon \mathbb{R} \to \mathbb{R}$ be a function satisfying $\sigma(x + c) = \sigma(x) + c$. Then $\sigma$ is a translation.*

*Proof.* Let $a = \sigma(0)$. Then $\sigma(x) = \sigma(x + c) - c$. Choosing $c = -x$ gives $\sigma(x) = a + x$. $\qquad\square$

**Proposition 10.** *Let $X$ and $K$ be as in Prop 8. Let $f$ be a convolutional layer with kernel $K$ and $\sigma$ an activation function. Assume $\sigma \colon \mathbb{R} \to \mathbb{R}$ is piecewise differentiable. Then if the composition $\varphi = \sigma \circ f$ is equivariant with respect to arbitrary uniform motions, it is an affine map of the form $\varphi(\boldsymbol{X}) = K' * \boldsymbol{X} + b$, where $b$ is a real number and $\sum_v K'(v) = 1$.*

*Proof.* If $f$ is non-zero, then we can choose a tensor $X$, and constant tensor $C$ full of $c \in \mathbb{R}$, and $p \in \mathbb{Z}^2$ such that $c$ and $\beta = (f(X))_p$ are any two real numbers. Let $\lambda = \sum_v K(v)$. As before $f(C) = \lambda C$. Equivariance thus implies

$$\sigma(\beta + c\lambda) = \sigma(\beta) + c.$$

Note $\lambda \neq 0$, since if $\lambda = 0$, then $\sigma(\beta) = \sigma(\beta) + c$ implies $c = 0$. However $c$ is arbitrary. Let $h = c\lambda$. Then

$$\frac{\sigma(\beta + h) - \sigma(\beta)}{h} = \frac{1}{\lambda}.$$

This holds for arbitrary $\beta$ and $h$, and thus we find $\sigma$ is everywhere differentiable with slope $\lambda^{-1}$. So $\sigma(x) = x/\lambda + b$ for some $b \in \mathbb{R}$. We can then rescale the convolution kernel $K' = K/\lambda$ to get $\varphi(\boldsymbol{X}) = K' * \boldsymbol{X} + b$. $\qquad\square$

**Corollary 11** (Corollary 2). *If $f$ is a CNN alternating between convolutions $f_i$ and pointwise activations $\sigma_i$ and the combined layers $\sigma_i \circ f_i$ are uniform motion equivariant, then $f$ is affine.*

*Proof.* This follows from Proposition 9 and the fact that composition of affine functions is affine. $\quad\square$

Since our treatment is only for pointwise activation functions, it remains a possibility that more descriptive networks can be constructed using activation functions which span multiple channels.

**Proposition 12** (Proposition 3). *A residual block $f(\boldsymbol{x}) + \boldsymbol{x}$ is uniform motion equivariant if the residual connection $f$ is uniform motion invariant.*

*Proof.* We denote the uniform motion transformation by $\boldsymbol{c}$ by $T_{\boldsymbol{c}}^{\mathrm{um}}(\boldsymbol{w}) = \boldsymbol{w} + \boldsymbol{c}$. Let $f$ be an invariant residual connection which is a composition of convolution layers and activation functions. Then we compute

$$
\begin{aligned}
f(T_{\boldsymbol{c}}^{\mathrm{um}}(\boldsymbol{w})) + T_{\boldsymbol{c}}^{\mathrm{um}}(\boldsymbol{w}) &= f(\boldsymbol{w}) + \boldsymbol{w} + \boldsymbol{c} \\
&= (f(\boldsymbol{w}) + \boldsymbol{w}) + \boldsymbol{c} \\
&= T_{\boldsymbol{c}}^{\mathrm{um}}(f(\boldsymbol{w}) + \boldsymbol{w}).
\end{aligned}
$$

as desired. □

### B.4 RESULTS ON SCALE EQUIVARIANCE

We show that a scale-invariant CNN in the sense of equation 1 would be extremely limited. Let $G = (\mathbb{R}_{>0}, \cdot)$ be the rescaling group. It is isomorphic to $(\mathbb{R}, +)$. For $c$ a real number, $\rho_c(\lambda) = \lambda^c$ gives an action of $G$ on $\mathbb{R}$. There is also, e.g., a two-dimensional representation

$$
\rho(\lambda) = \begin{pmatrix} 1 & \log(\lambda) \\ 0 & 1 \end{pmatrix}.
$$

**Proposition 13.** *Let $K$ be a $G$-equivariant kernel for a convolutional layer. Assume $G$ acts on the input layer by $\rho_{in}$ and output layer by $\rho_{out}$. Assume that the input layer is padded with 0s. Then $K$ is 1x1.*

*Proof.* If $v \neq 0$ then there exists $\lambda \in \mathbb{R}_{>0}$ such that $\lambda v$ is outside the radius of the kernel. So $K(\lambda v) = 0$. Thus by equivariance, for some $n$,

$$
K(v) = \lambda^{\mathbf{n}} \rho_{\mathrm{out}}^{-1} K(\lambda v) \rho_{\mathrm{in}} = 0.
$$

□

### B.5 EQUIVARIANCE ERROR.

In practice it is difficult to implement a model which is perfectly equivariant. This results in equivariance error $\mathrm{EE}_T(x) = |T(f(x)) - f(T(x))|$. Given an input $x$ with true output $\hat{y}$ and transformed data $T(x)$, the transformed test error $\mathrm{TTE} = |T(\hat{y}) - f(T(x))|$ can be bounded using the untransformed test error $\mathrm{TE} = |\hat{y} - f(x)|$ and EE.

**Proposition 14.** *The transformed test error is bounded*

$$
\mathrm{TTE} \leq |T|\mathrm{TE} + \mathrm{EE}. \tag{6}
$$

*Proof.* By the triangle inequality

$$
\begin{aligned}
|T(\hat{y}) - f(T(x))| &\leq |T(\hat{y}) - T(f(x))| + |T(f(x)) - f(T(x))| \\
&= |T||\hat{y} - f(x)| + \mathrm{EE}.
\end{aligned}
$$

□

For uniform motion $\mathrm{TTE} \leq \mathrm{EE} + \mathrm{TE}$ since $|T(\hat{y}) - T(f(x))| = |\hat{y} + c - f(x) - c| = \mathrm{TE}$. Consider $x$ and $y$ as flattened into a vector. $|T| = \sup_{|x|=1} |T(x)|$ denotes the operator norm. For $g \in SO(2)$, acting by $T_g$ on vector fields, $|T_g| = 1$. For scaling $T^\lambda(w)(x,t) = \lambda w(\lambda x, \lambda^2 t)$, $|T^\lambda| = \lambda/\sqrt{\lambda^4} = 1/\lambda$.

### B.6 FULL LISTS OF SYMMETRIES OF HEAT AND NS EQUATIONS.

**Symmetries of NS Equations.** The Navier-Stokes equations are invariant under five different transformations (see e.g. [38]),

- Space translation: $T_{\boldsymbol{c}}^{\text{sp}} \boldsymbol{w}(\boldsymbol{x}, t) = \boldsymbol{w}(\boldsymbol{x} - \boldsymbol{c}, t)$, $\boldsymbol{c} \in \mathbb{R}^2$,
- Time translation: $T_{\tau}^{\text{time}} \boldsymbol{w}(\boldsymbol{x}, t) = \boldsymbol{w}(\boldsymbol{x}, t - \tau)$, $\tau \in \mathbb{R}$,
- Uniform motion: $T_{\boldsymbol{c}}^{\text{um}} \boldsymbol{w}(\boldsymbol{x}, t) = \boldsymbol{w}(\boldsymbol{x}, t) + \boldsymbol{c}$, $\boldsymbol{c} \in \mathbb{R}^2$,
- Reflect/rotation: $T_R^{\text{rot}} \boldsymbol{w}(\boldsymbol{x}, t) = R \boldsymbol{w}(R^{-1}\boldsymbol{x}, t)$, $R \in O(2)$,
- Scaling: $T_{\lambda}^{sc} \boldsymbol{w}(\boldsymbol{x}, t) = \lambda \boldsymbol{w}(\lambda \boldsymbol{x}, \lambda^2 t)$, $\lambda \in \mathbb{R}_{>0}$.

Individually each of these types of transformations generates a group of symmetries of the system. Collectively, they form a 7-dimensional symmetry group.

**Symmetries of Heat Equation.** The heat equation has an even larger symmetry group than the NS equations [38]. Let $H(\boldsymbol{x}, t)$ be a solution to equation $\mathcal{D}_{\text{heat}}$. Then the following are also solutions:

- Space translation: $H(\boldsymbol{x} - \boldsymbol{v}, t)$, $\boldsymbol{v} \in \mathbb{R}^2$,
- Time translation: $H(\boldsymbol{x}, t - c)$, $c \in \mathbb{R}$,
- Galilean: $e^{-\boldsymbol{v}\cdot\boldsymbol{x} + \boldsymbol{v}\cdot\boldsymbol{v}t} H(x - 2\boldsymbol{v}t, t)$, $\boldsymbol{v} \in \mathbb{R}^2$
- Reflect/Rotation: $H(R\boldsymbol{x}, t)$, $R \in O(2)$,
- Scaling: $H(\lambda\boldsymbol{x}, \lambda^2 t)$, $\lambda \in \mathbb{R}_{>0}$
- Linearity: $\lambda H(\boldsymbol{x}, t)$, $\lambda \in \mathbb{R}$ and $H(\boldsymbol{x}, t) + H_1(\boldsymbol{x}, t)$, $H_1 \in \text{Sol}(\mathcal{D}_{\text{heat}})$
- Inversion: $a(t)e^{-a(t)c\boldsymbol{x}\cdot\boldsymbol{x}} H(a(t)\boldsymbol{x}, a(t)t)$, where $a(t) = (1 + 4ct)^{-1}$, $c \in \mathbb{R}$.

### B.7 TURBULENCE KINETIC ENERGY SPECTRUM

The turbulence kinetic energy spectrum $E(k)$ is related to the mean turbulence kinetic energy as

$$\int_0^{\infty} E(k)dk = (\overline{(u')^2} + \overline{(v')^2})/2,$$

$$\overline{(u')^2} = \frac{1}{T} \sum_{t=0}^{T} (u(t) - \bar{u})^2,$$

where the $k$ is the wavenumber and $t$ is the time step. Figure 5 shows a theoretical turbulence kinetic energy spectrum plot. The spectrum can describe the transfer of energy from large scales of motion to the small scales and provides a representation of the dependence of energy on frequency. Thus, the Energy Spectrum Error can indicate whether the predictions preserve the correct statistical distribution and obey the energy conservation law. A trivial example that can illustrate why we need ESE is that if a model simply outputs moving averages of input frames, the accumulated RMSE of predictions might not be high but the ESE would be really big because all the small or even medium eddies are smoothed out.

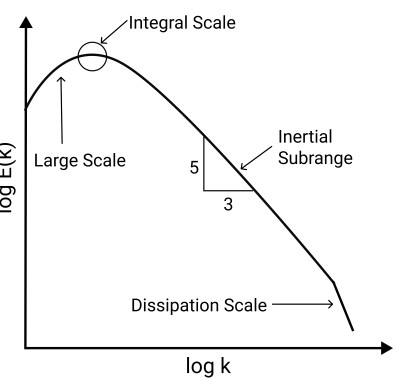

Figure 5: Theoretical turbulence energy spectrum plot

## C HEAT DIFFUSION

**2D Heat Equation.** Let $H(t, x, y)$ be a scalar field representing temperature. Then $H$ satisfies

$$\frac{\partial H}{\partial t} = \alpha \Delta H. \tag{$\mathcal{D}_{\text{heat}}$}$$

Here $\Delta = \partial_x^2 + \partial_y^2$ is the two-dimensional Laplacian and $\alpha \in \mathbb{R}_{>0}$ is the diffusivity.

The Heat Equation plays a major role in studying heat transfer, Brownian motion and particle diffusion. We simulate the heat equation at various initial conditions and thermal diffusivity using the finite difference method and generate $6k$ scalar temperature fields. Figure 6 shows a heat diffusion process where the temperature inside the circle is higher than the outside and the thermal diffusivity is 4. Since the heat equation is much simpler than the NS equations, a shallow `CNN` suffices to forecast the heat diffusion process.

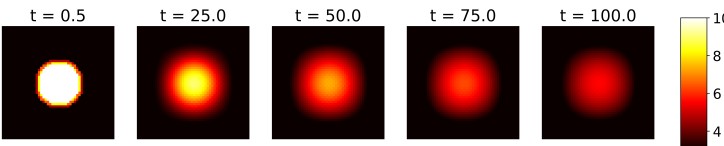

Figure 6: Five snapshots in heat diffusion dynamics. The spatial resolution is 50×50 pixels.

For heat diffusion, due to the law of energy conservation, the sum of each temperature field should be consistent over the entire heat diffusion process. We evaluate the physical characteristics of the predictions using the L1 loss of the thermal energy. Table 5 shows the prediction RMSE and thermal energy loss of the `CNNs` and three `Equ-CNNs` on three transformed test sets. We can see that `Equ-CNNs` consistently outperform `CNNs` over three test sets.

Table 5: The prediction RMSE and thermal energy L1 loss of the `CNNs` and three `Equ-CNNs` on three **transformed** test sets. `Equ-CNNs` outperform the `CNNs` over all three test sets.

| Testsets | RMSE (Thermal Energy Loss) | | |
| Models | *Mag* | *Rot* | *Scale* |
|---|---|---|---|
| `CNNs` | 0.103 (4696.3) | 0.308 (1125.6) | 0.357 (1447.6) |
| `Equ-CNNs` | **0.028 (107.7)** | **0.153 (127.3)** | **0.045 (396.6)** |

## D  IMPLEMENTATION DETAILS

### D.1  DATASETS DESCRIPTION

**Rayleigh-Bénard convection**  Rayleigh-Bénard convection results from a horizontal layer of fluid heated from below, which is a major feature of the El Nino dynamics. The dataset comes from two dimensional turbulent flow simulated using the Lattice Boltzmann Method [8] with Rayleigh number $= 2.5 \times 10^8$. We divided each $1792 \times 256$ image into 7 square sub-regions of size $256 \times 256$, then downsample them into $64 \times 64$ pixels sized images. Figure 7 in appendix shows a snapshot in our RBC flow dataset. We generate the following test sets to test the models' generalization ability.

- *Uniform motion (UM)*: transformed test sets by adding random vectors drawn from $U(-1, 1)$.

- *Magnitude (Mag)*: transformed test sets by multiplying random values sampled from $U(0, 2)$.

- *Rotation (Rot)*: transformed test sets by randomly rotated by the multiples of $\pi/12$.

- *Scale*: transformed test sets by scaling each sample $\lambda$ sampled from $U(1/5, 2)$.

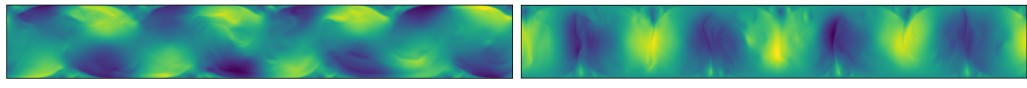

Figure 7: A snapshot of the Rayleigh-Bénard convection flow, the velocity fields along $x$ direction (left) and $y$ direction (right) [8]. The spatial resolution is 1792×256 pixels.

**Ocean Currents**   We used the reanalysis ocean currents velocity data generated by the NEMO (Nucleus for European Modeling of the Ocean) simulation engine [2]. We selected an area from each of the Atlantic, Indian and North Pacific Oceans from 01/01/2016 to 08/18/2017 and extracted $64\times64$ sub-regions for our experiments. The corresponding latitude and longitude ranges for the selected regions are (-44∼-23, 25∼46), (55∼76, -39∼-18) and (-174∼-153, 5∼26) respectively. We not only test all models on the future data but also on a different domain (-180∼-159, -40∼-59) in South Pacific Ocean from 01/01/2016 to 12/15/2016. Also, the most recent work on this dataset is [15], which unified a warping scheme and an U-net to predict temperature. So to compare our equivariant models with state-of-arts, we also investigate our models on the task of temperature field predictions. Since the data back to year 2006 that [15] used is no longer available, we collect more recent temperature data from a square region (-50∼-20, 20∼50) in Atlantic Ocean from 01/01/2016 to 12/31/2017.

## D.2   EXPERIMENTS SETUP

We tested our convolutional equivariant layers in two architecture, 18-layer `ResNet` and 13-layer `U-net`. One of our goals is to show that adding equivariance improves the physical accuracy of state-of-the-art dynamics prediction. `ResNet` and `U-net` are the popular state-of-the-art methods at the moment and our equivariance techniques are well-suited for their architecture. The reason we did not use recurrent models, such as Convolutional LSTM, is that they are slow to train especially for our case where the input length is large. This does not fit our long-term goal of accelerating computation.

The input to each model is a $l \times 64 \times 64 \times 2$-size tensor representing the past $l$ timesteps of the velocity field. The output is a single velocity field. The value of $l$ is a hyper-parameter we tuned. We found the optimal value of $l$ to be around $l = 25$. To predict more timesteps, we apply the model autoregressively, dropping the oldest timestep and concatenating the prediction to the input.

To make this a fair comparison, we adjust the hidden dimensions for different equivariant models to make sure that the number of parameters in all models are about the same for either architecture, which can be found in Table 6. Table 7 gives the hyper-parameter tuning ranges for our models. Note that the hidden dimension and the number of layers of the shallow CNNs for the heat diffusion task are also well-tuned.

The loss function used is the MSE between the predicted frames and the ground truth for next $k$ steps, where $k$ is a parameter we tuned. We found $k = 3$ or $4$ give the best performance. We use 60%-20%-20% training-validation-test split in time and use the validation set for hyper-parameters tuning based on the average error of predictions. The training set corresponds to the first 60% of the entire dataset in time and the validation/test sets contains the following 40%. For fluid flows, we standardize the data by the average of velocity vectors and the standard deviation of the L2 norm of velocity vectors. For sea surface temperature, we did the exact same data preprocessing described in de Bezenac et al. [15].

Table 6: The number of parameters in each model and time costs for training an epoch on 8 V100 GPUs.

| **ResNet** | *Reg* | *UM* | *Mag* | *Rot* | *Scale* | **U-net** | *Reg* | *UM* | *Mag* | *Rot* | *Scale* |
|---|---|---|---|---|---|---|---|---|---|---|---|
| Params ($10^6$) | 11.0 | 11.0 | 11.0 | 10.2 | 10.7 | | 6.2 | 6.2 | 6.2 | 7.1 | 5.9 |
| *Time(min)* | 3.04 | 5.21 | 5.50 | 14.31 | 160.32 | | 2.15 | 4.32 | 4.81 | 11.32 | 135.72 |

## E   ADDITIONAL RESULTS

Table 8 shows the RMSEs of temperature predictions. Figure 8 shows the ground truth and the predicted velocity norm fields ($\sqrt{u^2 + v^2}$) at time step 1, 5 and 10 by the `U-net` and four `Equ-Unet`

---

[2]The data are available at `https://resources.marine.copernicus.eu/?option=com_csw&view=details&product_id=GLOBAL_ANALYSIS_FORECAST_PHY_001_024`

Table 7: The Hyper-parameter tuning range: Learning rate, the number of accumulated errors for backpropogation, the number of input frames, batch size, and the hidden dimension and the number of layers of the shallow CNNs for heat diffusion

| Learning rate | #Accum Errors | #Input frames | Batch Size | Hidden dim (CNNs) | #Layers (CNNs) |
|---|---|---|---|---|---|
| 1e-1 ∼ 1e-6 | 1∼10 | 1∼30 | 4∼64 | 8∼128 | 1∼10 |

on the four transformed test samples. Figure 9 shows the ground truth and the predicted ocean currents ($\sqrt{u^2 + v^2}$) at time step 5 and 10 by the regular `ResNet` and four `Equ-ResNets` on the test set of future time.

Table 8: The RMSEs of temperature predictions on test data. For equivariant models, the left number in the cell is `ResNet` and the right number in the cell is `U-net`

| | CLSTM | Bézenac | ResNet | U-net | Equ$_{UM}$ | Equ$_{Mag}$ | Equ$_{Rot}$ | Equ$_{Scal}$ |
|---|---|---|---|---|---|---|---|---|
| RMSE | 0.46 | 0.38 | 0.41 | 0.391 | 0.38 \| **0.37** | 0.39 \| 0.37 | 0.38 \| 0.40 | 0.42 \| 0.41 |

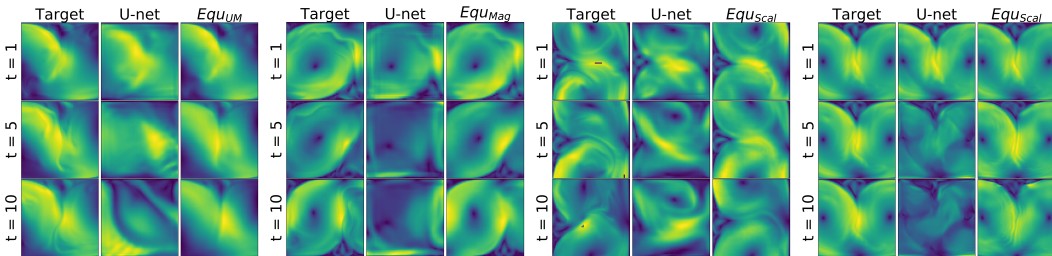

Figure 8: The ground truth and the predicted velocity norm fields ($\sqrt{u^2 + v^2}$) at time step 1, 5 and 10 by the `U-net` and four `Equ-Unet` on the four transformed test samples. From left to right, the transformed test samples are the original test samples uniform-motion-shifted by $(1, -0.5)$, magnitude-scaled by 1.5, rotated by 90 degrees and upscaled by 3 respectively. The first row is the target, the second row is `Equ-Unets` predictions, and the third row is predictions by `U-net`.

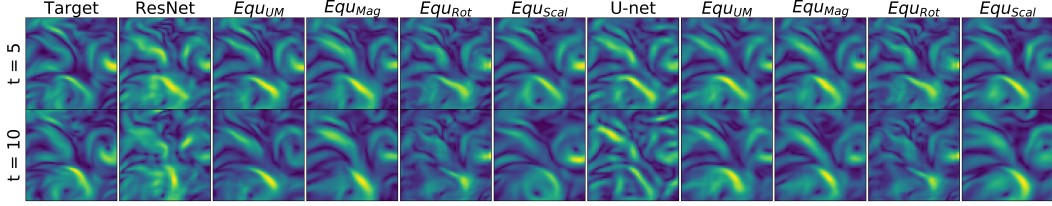

Figure 9: The ground truth and the predicted ocean currents ($\sqrt{u^2 + v^2}$) at time step 5 and 10 by the regular `ResNet` and four `Equ-ResNets` on the test set of future time.

