# OpenReview forum: "Incorporating Symmetry into Deep Dynamics Models for Improved Generalization"
_ICLR.cc/2021/Conference — ICLR 2021 Poster_

### Official Review · AnonReviewer3 · 2020-10-28
**Fluid dynamics models that incorporate physical symmetries generalize well under distribution shift, but the role of equivariance isn't yet directly validated.**

**Rating:** 7
**Confidence:** 3

**Review:**

Summary: The paper demonstrates that incorporating equivariance (i.e. symmetries) into model for predicting fluid dynamics improves its performance, especially when the test distribution is transformed by those symmetry groups. Leveraging the recent literature on equivariant CNNs, the paper proposes a CNN model that is equivariant with respect to known symmetries of the Navier-Stokes equations (time/space translation, rotation, uniform motion, and scaling). This approached is validated on 2 datasets on fluid dynamics: a synthetic dataset on Rayleigh-Benard convection and a real-world ocean dynamics dataset. On the synthetic dataset, the proposed models demonstrate better performance under distribution shift. On the real-world dataset, the models yields predictions that are more accurate and physically consistent.

======

Strengths:
1. The problem setting and datasets are realistic and could have high impact. Given the growing importance of machine learning in physical dynamics and climate science, better understanding on how to incorporate prior knowledge (symmetries) could lead to better modeling.
2. The claim of generalization under distribution shift is well-validated. The paper identifies 5 types of symmetries of Navier-Stokes equations, and in the synthetic dataset perturbs the test set by applying transforms from these symmetry groups. The experiment results show that models that incorporate those symmetries perform better on this perturbed dataset (though it's unclear how a model that incorporates one symmetry, e.g. scale symmetry, performs on test set that was transformed by another symmetry, e.g., rotation.)

Weaknesses:
1. The claim of model equivariance is not directly validated. Even though the main claim that incorporating equivariance leads to better generalization, model equivariance is never measured directly. There could be components of the model that are not equivariant, such as pooling or convolution with stride 2. At least the U-net architecture uses stride 2, so it might not be equivariance. It would be interesting if the degree of equivariance could be measured, to see how that relates to the generalization ability.
2. Sample complexity claim is not validated. In the abstract, the proposed models are claim to "enjoy favorable sample complexity". However, there is no experiment validating this claim. For example, with varying amount of data, how do the models perform?
3. The need for separate models for each of the different symmetries. For each of rotation, uniform motion, and scale symmetries, there is a model that incorporates such symmetry. However, it's unclear if these could be combined. In the conclusion, it is claimed that "all of our equivariant models can be combined". But the conclusion also states that "there does not exist a single model with equivariance to the full symmetry group of the Navier-Stokes equation".
4. [Minor] Limited novelty. It's not clear how the proposed models different from equivariant CNNs in the literature [9, 10, 11, 47]. For symmetries not covered in existing work (e.g. uniform motion), the proposed equivariant model seems ad-hoc. However, I do recognize that making all existing architectures work for this realistic problem is non-trivial, so this is only a minor weakness.

======

Overall, I'm on the fence about accepting/rejecting. The proposed model is shown to generalize well under specific distribution shift, but the main claim about the role of equivariance is not yet validated directly. Addressing this concern would make the paper stronger.

======

Additional feedback and questions:
- Visualization in Figure 1 is great.
- The intro mentions Noether's theorem about the correspondence between symmetry and conserved quantity. Is there any evidence that the proposed models conserve some quantity?

====
After rebuttal: Thank you for addressing my concerns.
- I believe the claim of equivariance leading to improved generalization (weakness 1 above) has now been validated more directly.
- Regarding the validation of sample complexity claim: a more direct investigation of performance against training set size, for equivariant and non-equivariant models, would be more convincing. I understand that augmenting data does increase the training set size by a factor of 3, but the training set is then not iid so it's not clear what the "effective size" of training set is. In any case I do believe that equivariance reduces sample complexity.

Overall, I vote for accepting.

---

> ### Author Response · Authors · 2020-11-20
> **Response to Reviewer3: Part One**
>
> We are glad you feel our problem is high-impact and that our methods could lead to better modeling.   Thank you for your constructive questions. We agree with the suggestion that a more thorough validation of equivariance will strengthen our paper, and so we have done additional experiments and revised our paper accordingly.
>
> $\textbf{Q1:}$ The claim of model equivariance is not directly validated. Even though the main claim that incorporating equivariance leads to better generalization, model equivariance is never measured directly. There could be components of the model that are not equivariant, such as pooling or convolution with stride 2. At least the U-net architecture uses stride 2, so it might not be equivariance. It would be interesting if the degree of equivariance could be measured, to see how that relates to the generalization ability.
>
> $\textbf{Answer:}$
> This is a good point.  We have empirically tested the equivariance of our models and added this to the paper (see Table 4.)  We can use equivariance error as described in Appendix E.6 to measure the equivariance of our models.  As expected the uniform motion and magnitude equivariant models have zero equivariance error.  The rotation and scaling equivariant models have some equivariance error, but much less than the non-equivariant models.  This makes sense as both of these models approximate the symmetry group with a discrete version.
>
> We may think of stride as a subsampling operation which is performed after convolution.  Thus since the composition of equivariant operations is equivariant, convolution with stride is equivariant if both convolution and stride individually are equivariant.
> It is true that stride interacts with the equivariance of the model to translations.  A convolution with stride $n$ is translation equivariant between two layers $L_1 \to L_2$ in which the translation group acts by shifts of size $n$ in $L_1$ and size 1 in $L_2$.  Thus stride makes the network only translation equivariant to larger shifts.  However, this is not an issue for uniform motion, magnitude, rotation, or scale equivariance.  All of these symmetry group actions commute with the striding operation.
>
> $\textbf{Q2:}$ Sample complexity claim is not validated. In the abstract, the proposed models are claim to "enjoy favorable sample complexity". However, there is no experiment validating this claim. For example, with varying amount of data, how do the models perform?
>
> $\textbf{Answer:}$
> This claim is supported by our comparisons to training baselines with augmented data.  In these cases the baselines were trained on 3 times the data of the equivariant models and still reach similar performance in RMSE or ESE.  This comparison is in Table 1 for Raleigh-Benard convection.  To strengthen our claim we have added comparisons to data augmented baselines for the ocean dataset as well.  In the revised version of Table 3, one can see that even training on 3 times as much data, the augmented models fail to reach the performance of the equivariant models, and, in fact, perform worse than unaugmented baselines in ESE.
>
> In Section 5.2, we compared ResNet and Equ-ResNet when both train and test sets have random transformations from the relevant symmetry group applied to each sample. Table 2 shows Equ-ResNet outperforms ResNet on average by 34% RMSE and 40% ESE, which demonstrates that Equ-ResNet is more sample efficient.
>
> Lastly, we also theoretically proved equivariant networks are better than the techniques of data augmentation in Section 3.1 and Appendix B.1.
>
> $\textbf{Q3:}$ The need for separate models for each of the different symmetries. For each of rotation, uniform motion, and scale symmetries, there is a model that incorporates such symmetry. However, it's unclear if these could be combined. In the conclusion, it is claimed that "all of our equivariant models can be combined". But the conclusion also states that "there does not exist a single model with equivariance to the full symmetry group of the Navier-Stokes equation".
>
> $\textbf{Answer:}$
> We agree it would be best to combine the symmetries into a single model.   It is possible, but doing so is non-trivial, and we continue to work on the problem.  In conclusion, we point out such a combined model has not yet been created.  The challenge stems from the varied types of symmetries included: linear, affine, compact and non-compact.  In light of this, our work demonstrating translation can be combined with each of other symmetries and that this improves performance is already a significant step.

---

> > ### Author Response · Authors · 2020-11-20
> > **Response to Reviewer3: Part Two**
> >
> > $\textbf{Q4:}$ [Minor] Limited novelty. It's not clear how the proposed models different from equivariant CNNs in the literature. For symmetries not covered in existing work (e.g. uniform motion), the proposed equivariant model seems ad-hoc. However, I do recognize that making all existing architectures work for this realistic problem is non-trivial, so this is only a minor weakness.
> >
> > $\textbf{Answer:}$
> > We agree that making these methods work on realistic problems is quite challenging.
> > For rotational equivariance, we do use the methods of [9,10,49], however, our main methodological contributions consist of non-trivial adaptations to scale equivariance and our effective implementation of magnitude and uniform motion equivariance (3.4,3.5).  We disagree our method for uniform motion is ad-hoc.  It is theoretically supported by Prop 3 and Appendix B.3.  This method may be thought of conjugation of the dot product operation by a canonicalization [C1, C2, C3].  That is, we first shift the input into a canonical form, then apply the operation, and then shift the output back.
> >
> > Our implementation of scale equivariance differs significantly from that of Worrall&Welling [51] to account for the differences in a physical system:
> > Our implementation of group correlation directly incorporates the physical scaling law of the NS system. We scale the input of every layer anisotropically, i.e. differently across time and space and magnitude.
> > Our model uses antialiased rescaling to achieve equivariance to both up and down scaling by any real number, not just powers of two and not just downscaling, as in [51].
> > We treat the channels as the time axis, and we increase the dimension of this axis with layer depth.  This avoids convolving across the time axis, allowing us to use conv2D instead of conv3D,  which is computationally expensive.
> >
> > $\textbf{Q5:}$ The intro mentions Noether's theorem about the correspondence between symmetry and conserved quantity. Is there any evidence that the proposed models conserve some quantity?
> >
> > $\textbf{Answer:}$
> > Good question.  The Rayleigh-Benard convection and real ocean data are not perfectly energy preserving since they do not represent closed systems.  Energy may be lost or gained from heat or flux along the boundary.  That said, we can measure energy spectrum error which shows whether the model correctly predicts the amount of energy lost or gained correctly.  To do well on this metric, the model must conserve energy where applicable.  In fact, this metric is even more refined since it tracks how much energy exists at different scales, and so doing well on this metric requires the model to correctly track energy between different scales.  Equivariance drastically improves the performance of our models on this metric.
> >
> > In the case of the heat diffusion experiment considered in the appendix, we have a closed system and so measuring energy error with the ground truth is equivalent to measuring energy conservation.  We find the equivariant models conserve energy much better as shown in table 5 in Appendix C.
> >
> > $\textbf{Q6:}$ The proposed model is shown to generalize well under specific distribution shift, but the main claim about the role of equivariance is not yet validated directly.
> >
> > $\textbf{Answer:}$
> > Our models and baselines are very similar up to replacing convolutions with equivariant convolutions.  Thus it seems the drastic improvement in accuracy and energy spectrum error can be attributed to equivariance.
> >
> > [C1] Geoffrey E. Hinton. A Parallel Computation that Assigns Canonical Object-Based Frames of Reference, IJCAI, 1981.
> >
> > [C2] Yangyan Li, Rui Bu, Mingchao Sun, Wei Wu, Xinhan Di, Baoquan Chen. PointCNN: Convolution On X -Transformed Points.32nd Conference on Neural Information Processing Systems (NeurIPS 2018)
> >
> > [C3] Max Jaderberg, Karen Simonyan, Andrew Zisserman, Koray Kavukcuoglu. Spatial Transformer Networks. Advances in Neural Information Processing Systems 28 (NIPS 2015)

---

### Official Review · AnonReviewer4 · 2020-10-29
**Redesign of CNN to make it symmetric as physical dynamics**

**Rating:** 4
**Confidence:** 4

**Review:**

Physical dynamics have symmetry properties, which can be leveraged by neural networks for better accuracy and generalization. This paper takes 2D Navier-Stokes (NS) equation as an example and re-design the convolutions in networks. Simulations and experiments on real data are conducted to verify that the new models are effective.

The common logic of the work is reasonable. However, the novelty and technical contribution is limited. 1) the symmetries of NS equations are well-studied and borrowed from [37]. 2) The design of Equivariant networks, i.e., Equivariant ResNet and U-Net are incremental comparing to [9,10] and [47].

Some further questions: 1. Are the proposed re-design for uniform motion equivariance in eqn (2) and that for scale equivariance in eqn (5) compatible? Can we design networks that simultaneously have multiple kinds of symmetry? Even potentially to automatedly learn the symmetry of the unknown systems? It is more attractive to me than the current approach. 2. What is the detailed implementation of Augm in Table 1? How many augmentation for each instance on average? Will the performance increase with the number of augmentation? Is it possible to do Augm for Ocean dynamic? More experiments will firmly validate the superiority of the proposed method over the data augmentation approach. 3. Shall we keep convolution, which are original designed for image, in physical dynamics?

Sec 2.1-2.3 are hard to follow. Fortunately, I can get the main idea with almost skip this part. Maybe this part is not necessarily in that rigorous. Otherwise, please move some theoretical results into the body to highlight the technical contribution.

Overall, this paper proposed methods to make CNN to be symmetric as (NS equation) dynamics. The technical contribution need be highlighted.

---

> ### Author Response · Authors · 2020-11-10
> **Urgent  Issue: this review is for a different paper**
>
> This review is for a different paper

---

> > ### Comment · AnonReviewer4 · 2020-11-11
> > **Here is the right review**
> >
> > Thanks for the reminder. I put the right review. Sorry for that and the confusion.

---

> ### Author Response · Authors · 2020-11-20
> **Response to Reviewer 4: Part One**
>
> Thank you for your comments and questions. Your question (along with AnonReviewer2) about data augmentation for ocean dynamics motivated us to perform an additional experiment which shows strong results and we feel improves our paper further.
>
> Physical dynamics have symmetry properties, which can be leveraged by neural networks for better accuracy and generalization. This paper takes 2D Navier-Stokes (NS) equation as an example and re-design the convolutions in networks. Simulations and experiments on real data are conducted to verify that the new models are effective.
>
> $\textbf{Q1:}$ The common logic of the work is reasonable. However, the novelty and technical contribution is limited. 1) the symmetries of NS equations are well-studied and borrowed from [37].
>
> $\textbf{Answer:}$
> The purpose of our paper is to show how previous physical knowledge can be leveraged to improve neural network performance. Most existing equivariant neural networks are applied to either static images [49, 51] or texts [D4]. The main novelty of our work is to develop equivariant neural architecture for dynamical systems. In this sense, the fact that the symmetries of the NS equations are well-studied is part of our point.  We see equivariant neural networks as a way to make use of this knowledge instead of starting over with a purely data-driven approach.
>
>
> $\textbf{Q2:}$ The design of Equivariant networks, i.e., Equivariant ResNet and U-Net are incremental comparing to ...
>
> $\textbf{Answer:}$
> For rotational equivariance, we do use the methods of [49, 9,10], however, our main methodological contributions consist of non-trivial adaptations to scale equivariance and our effective implementation of magnitude and uniform motion equivariance (3.4,3.5).
>
> Our implementation of scale equivariance differs significantly from that of Worrall&Welling [51] to account for the differences in a physical system:
> Our implementation of group correlation directly incorporates the physical scaling law of the NS system. We scale the input of every layer anisotropically, i.e. differently across time and space and magnitude.
> Our model uses antialiased rescaling to achieve equivariance to both up and down scaling by any real number, not just powers of two and not just downscaling, as in [51].
> We treat the channels as the time axis, and we increase the dimension of this axis with layer depth.  This avoids convolving across the time axis, allowing us to use conv2D instead of conv3D,  which is computationally expensive.
>
> $\textbf{Q3:}$ Are the proposed re-design for uniform motion equivariance in eqn (2) and that for scale equivariance in eqn (5) compatible? Can we design networks that simultaneously have multiple kinds of symmetry?
>
> $\textbf{Answer:}$
> It is possible to combine the different equivariant models we consider, but doing so is non-trivial, and we continue to work on the problem.  To the best of our knowledge, there does not yet exist a single model with equivariance to the full symmetry group of the Navier-Stokes equations.  The challenge stems from the varied types of symmetries included: linear, affine, compact and non-compact.  In light of this, our work demonstrating translation can be combined with each of other symmetries and that this improves performance is already a significant step.
>
> $\textbf{Q4:}$ Even potentially to automatedly learn the symmetry of the unknown systems?
>
> $\textbf{Answer:}$
> This is an interesting direction with a few recent works but goes in a different direction from our main point, which is in finding ways to incorporate physical knowledge you already have into neural networks.

---

> > ### Author Response · Authors · 2020-11-20
> > **Response to Reviewer 4: Part Two**
> >
> > $\textbf{Q5:}$. What is the detailed implementation of Augm in Table 1? How many augmentation for each instance on average? Will the performance increase with the number of augmentation? Is it possible to do Augm for Ocean dynamic? More experiments will firmly validate the superiority of the proposed method over the data augmentation approach
> >
> > $\textbf{Answer:}$
> > The Augmented training set contains additional transformed samples, and is thus three times as big as the original training set. We have added more details about this in the updated version. The performance would increase at first then decrease with the amount of augmentation because of the capacity limitation of non-equivariant models. The amount of augmented data is optimized. We already theoretically showed that equivariant networks are more efficient than data augmentation techniques in Appendix B.
> >
> > In response to your question and that of AnonReviewer2, we decided to add data augmentation as a competing baseline for the ocean experiments as well.  Thank you for the suggestion, the results are very strong in favor of equivariance and, we believe, improve our paper.  See Table 3 in our revised paper for the update results.  In all cases equivariant models outperformed baselines trained with data augmentation.  We find data augmentation sometimes improves slightly on RMSE but not as much as the equivariant models. And, in fact, ESE is uniformly worse for models trained with data augmentation than even the baselines. In contrast, the equivariant models have much better ESE than the baselines with or without augmentation.
> >
> > $\textbf{Q6:}$. Shall we keep convolution, which are original designed for image, in physical dynamics?
> >
> > $\textbf{Answer:}$
> > Convolutions are very natural for physical dynamics, for example, Green’s functions use convolution for solving linear ODEs.  The discrete Laplacian (which appears in discretized NS) can be represented as a convolution operation,/ Many recent papers on learning physical dynamics use convolution. which inspires [D3] to use CNNs as part of solving NS equations.   [D2] use a combination of parameterized and non-parameterized equations to solve an advection-diffusion equation. [D1] find ResNet is essentially performing Euler’s method using discrete convolution operations.
> >
> > $\textbf{Q7:}$ Sec 2.1-2.3 are hard to follow. Fortunately, I can get the main idea with almost skip this part. Maybe this part is not necessarily in that rigorous. Otherwise, please move some theoretical results into the body to highlight the technical contribution.
> >
> > $\textbf{Answer:}$
> > Sec 2.1-2.3 do not present new theoretical results, but they do explain the theoretical basis for our method in a rigorous way.  That is, given a differential equation, one can analytically determine its invariance symmetry group.  The ground truth forward prediction is then equivariant to this symmetry.  We can thus model forward prediction using a class of equivariant functions.   We will clarify this in the paper.
> >
> > [D1] Yiping Lu, Aoxiao Zhong, Quanzheng Li, and Bin Dong "Beyond finite layer neural networks: Bridging deep architectures and numerical differential equations." International Conference on Machine Learning. PMLR, 2018.
> >
> > [D2] Emmanuel de Bezenac,, Arthur Pajot, and Patrick Gallinari. "Deep learning for physical processes: Incorporating prior scientific knowledge." Journal of Statistical Mechanics: Theory and Experiment 2019.12 (2019): 124009.
> >
> > [D3] Jonathan Tompson, Kristofer Schlachter, Pablo Sprechmann, and Ken Perlin "Accelerating eulerian fluid simulation with convolutional networks." International Conference on Machine Learning. PMLR, 2017.
> >
> > [D4] Jonathan Gordon, David Lopez-Paz, Marco Baroni, Diane Bouchacourt. Permutation Equivariant Models for Compositional Generalization in Language. ICLR 2020.

---

### Official Review · AnonReviewer2 · 2020-11-01
**Novelty in modeling physical dynamics with symmetry**

**Rating:** 6
**Confidence:** 2

**Review:**

This paper studies improving the modeling of physical dynamics with equivariant neural networks. In particular, this paper focuses on a new type of data governed by physical models. Several special symmetry groups are considered to better characterize the system, including uniform motion equivariance, resolution-independent scaling, and resolution-dependent scaling, etc. Simulation results show that the proposed equivariant model yields better accuracy and physical consistency than the non-equivariant models even with data augmentation, given the type of distributional shift is known. Results on the real-world data show some of the equivariant models can generalize better than the non-equivariant models.

Pros
* The idea of using equivariant networks in physical dynamics seems well-motivated. In cases global alignment is difficult and the distributional shift is unknown, improving generalization by  incorporating known symmetries seems to be a natural idea.

* Although the idea of equivariant networks has been proposed before, the proposed treatments tailored to the modeling physical dynamics are new.

Cons

* It is claimed the data is governed by the differential equation, which has several symmetry properties. However, how the "ResNet and U-net" networks are used to solve the dynamics prediction problem is missing from the main text. Maybe due to the same reason, the connections to the differential equations are unclear. This paper is not quite self-contained.

* The content is targeted to a narrow audience.

Questions:

- Is data augmentation available as a baseline for experiments in Table 3?

- It seems different kinds of symmetries are incorporated separately - not sure if this is a limitation. If a system is known to satisfy multiple symmetries, is it possible to incorporate all of them together in a network?

---

> ### Author Response · Authors · 2020-11-20
> **Response to Reviewer2**
>
> Thank you for your comments and suggestions.  In particular, we appreciate your pointing out that our work applying equivariant methods to physical dynamics is well-motivated and our tailored methods are new.  Your question (along with AnonReviewer4) about comparing to data augmentation for ocean data motivated us to perform an additional experiment which shows strong results and we feel improves our paper further.
>
> $\textbf{Q1:}$ It is claimed the data is governed by the differential equation... the connections to the differential equations are unclear. This paper is not quite self-contained.
>
> $\textbf{Answer:}$
> ResNet and U-Net are commonly used in learning physical dynamics [B6, B7, B5]. Such layers could work in different types of convolutional networks.  We demonstrate this in two different convNets, U-Net and ResNet, which are popular and known to be effective in this domain.   We will add more description to the text.
>
> ResNet [B1] and U-net [B2] are compositions of many convolutional layers with skip connections and well-suited for our equivariance techniques. The input is the stacked historic frames and the output is the next frame. We incorporate the symmetries of NS equations instead of the equations themself, so the equivariant models we proposed can be applied to many other systems that have the same symmetry properties.
>
> $\textbf{Q2:}$ The content is targeted to a narrow audience.
>
> $\textbf{Answer:}$
> We respectfully disagree.  Dynamical system is ubiquitous in vision, robotics and physical sciences.Many recent works have proposed using DL to understand physical dynamics. See, for example, the survey papers, [B3, B4, B5] and many works cited within. Quoting AnonReviewer3: “Given the growing importance of machine learning in physical dynamics and climate science, better understanding on how to incorporate prior knowledge (symmetries) could lead to better modeling.”  Our contribution is to show how physics priors can be incorporated into NNs in the form of equivariance in real-world domains.
>
> $\textbf{Q3:}$ Is data augmentation available as a baseline for experiments in Table 3?
>
> $\textbf{Answer:}$
> We have added data augmentation as a baseline for the ocean experiments.  See Table 3 in our revised paper for the update results.  In all cases, equivariant models outperformed baselines trained with data augmentation.  We find data augmentation sometimes improves slightly on RMSE but not as much as the equivariant models. And, in fact, ESE is uniformly worse for models trained with data augmentation than even the baselines. In contrast, the equivariant models have much better ESE than the baselines with or without augmentation.
>
> We believe data augmentation presents a trade-off in learning.  Though the model may be less sensitive to the various transformations we consider, we need to train longer on many more samples.  The model may not have enough complexity to learn equivariance and the details of the fluid simulation at the same time.  In contrast, equivariant architectures do not have this trade-off.
>
> $\textbf{Q4:}$ It seems different kinds of symmetries are incorporated separately - not sure if this is a limitation. If a system is known to satisfy multiple symmetries, is it possible to incorporate all of them together in a network?
>
> $\textbf{Answer:}$
> Yes, It is possible to combine the different equivariant models, but doing so is non-trivial.  To the best of our knowledge, there does not yet exist a single model with equivariance to the full symmetry group of the Navier-Stokes equations.  The challenge stems from the varied types of symmetries included: linear, affine, compact and non-compact.  In light of this, our work demonstrating translation can be combined with each of other symmetries and that this improves performance is already a significant step.
>
> [B1] Kaiming He, Xiangyu Zhang, Shaoqing Ren, Jian Sun. Deep Residual Learning for Image Recognition. arXiv:1505.04597, 2015.
>
> [B2] Olaf Ronneberger, Philipp Fischer, Thomas Brox, U-Net: Convolutional Networks for Biomedical Image Segmentation. arXiv:1512.03385, 2015.
>
> [B3] Jared Willard, Xiaowei Jia, Shaoming Xu, Michael Steinbach, and Vipin Kumar. Integrating physics-based modeling with machine learning: A survey. arXiv:2003.04919, 2020.
>
> [B4] Jiequn Han, and Linfeng Zhang. "Integrating Machine Learning with Physics-Based Modeling." arXiv preprint arXiv:2006.02619 (2020).
>
> [B5] Steven L. Brunton, Bernd R. Noack, and Petros Koumoutsakos. "Machine learning for fluid mechanics." Annual Review of Fluid Mechanics 52 (2020): 477-508.
>
> [B6] Zongyi Li, Nikola Kovachki, Kamyar Azizzadenesheli, Burigede Liu, Kaushik Bhattacharya, Andrew Stuart, Anima Anandkumar. Fourier Neural Operator for Parametric Partial Differential Equations. arXiv: 2010.08895.
>
> [B7]  Rui Wang, Karthik Kashinath, Mustafa Mustafa, Adrian Albert, Rose Yu. Towards Physics-informed Deep Learning for Turbulent Flow Prediction. KDD 2020.

---

### Official Review · AnonReviewer1 · 2020-11-02
**an interesting problem but the presentation needs significant improvements**

**Rating:** 4
**Confidence:** 2

**Review:**

This work incorporates symmetries into a convolutional neural network to improve generalization performance and prediction accuracy. The work incorporates various symmetries by designing equivariant neural networks and demonstrate their superior performance on 2D time series prediction both theoretically and experimentally.

This work studies an interesting and important question in deep learning. However, the reviewer feels that the paper in the current form is difficult to follow with many places unclear. The overall result is based on a combination of previous work on equivariant convolutional neural network, the reviewer finds it hard to parse (the t and obtain a general methodology (or systematic approach) for dealing with symmetry from the work, and there is no high-level message conveyed in this work.

Below are some more detailed comments:

In Section 3.1, Equivariant Convolutions.: the result in (1) is quite unclear without much background provided. Equivariant ResNet and U-net: what is the implication on skip connection with no effects for network equivariance? Not clear.
For Section 3.2 - 3.5, the reviewer finds it very hard to follow the proposed approaches for dealing with each symmetry. (Perhaps this is because the reviewer has limited domain background in equivariant CNN) In general, the reviewer has it very hard to follow and interpret the proposed methods for dealing with each symmetry, and there is very little background provided.
3. It would be better if the author can provide more numerical results on real data other than simulated data. In Table 3, it seems that some of the methods with symmetry consideration are even worse than vanilla ResNet and U-net. This is a bit confusing.

Overall, the reviewer feels that the study is interesting with improved ways of dealing with symmetry for learning complex physical dynamics, but the work needs significant improvement in presentations.

---

> ### Author Response · Authors · 2020-11-20
> **Response to Reviewer1: Part One**
>
> Thank you for reviewing our work.  We are glad you found our study interesting and feel we have demonstrated our methods can improve the prediction of complex dynamics.  We address your specific questions and suggestions below.
>
> $\textbf{Q1:}$ The reviewer feels that the paper in the current form is difficult to follow with many places unclear. The overall result is based on a combination of previous work on equivariant convolutional neural network, the reviewer finds it hard to parse (the t and obtain a general methodology (or systematic approach) for dealing with symmetry from the work, and there is no high-level message conveyed in this work.
>
> $\textbf{Answer:}$
> In brief, our goal is to take advantage of first-principle physical knowledge to constrain the neural networks and improve their generalization.  Our general formulation is to take analytically determined symmetries of the differential equations governing a system and then use a network architecture which is equivariant to those symmetries to predict solutions.  The type of architecture which respects the symmetries varies with the type of symmetry group, compact or non-compact, affine or linear.  Thus we explore different architectures for different types of symmetries, described within Section 3.
>
> Our work is **NOT** simply a combination of previous work.  Applying scale-equivariance to high-dimensional real-world turbulence data is a difficult task.  Our methodological contributions consist of non-trivial adaptations to scale equivariance and our effective implementation of magnitude and uniform motion equivariance (3.4,3.5), which may be thought of conjugation of the dot product operation by a canonicalization [A1, A2, A3].  That is, we first shift the input into a canonical form, then apply the operation, and then shift the output back.
>
> Our implementation of scale equivariance differs significantly from that of Worrall & Welling [51] to account for the differences in a physical system:
> Our implementation of group correlation directly incorporates the physical scaling law of the NS system. We scale the input of every layer anisotropically, i.e. differently across time and space and magnitude.
> Our model uses antialiased rescaling to achieve equivariance to both up and down scaling by any real number, not just powers of two and not just downscaling, as in Worrall & Welling [51].
> We treat the channels as the time axis, and we increase the dimension of this axis with layer depth.  This avoids convolving across the time axis, allowing us to use conv2D instead of conv3D,  which is computationally expensive.
> We will edit the paper to make our approach clearer.
>
> $\textbf{Q2:}$ In Section 3.1, Equivariant Convolutions.: the result in (1) is quite unclear without much background provided.
>
> $\textbf{Answer:}$
> We have been brief since this is not our main contribution.  We will add additional background in the appendix and clarify this formula.
>
> $\textbf{Q3:}$ Equivariant ResNet and U-net: what is the implication on skip connection with no effects for network equivariance? Not clear.
>
> $\textbf{Answer:}$
> The implication is an equivariant neural network with skip connections is still fully equivariant.  Thus for models such as ResNet and U-net to be equivariant it is sufficient to use equivariant linear layers and equivariant activation functions.
>
>
> [A1] Geoffrey E. Hinton. A Parallel Computation that Assigns Canonical Object-Based Frames of Reference, IJCAI, 1981.
>
> [A2] Yangyan Li, Rui Bu, Mingchao Sun, Wei Wu, Xinhan Di, Baoquan Chen. PointCNN: Convolution On X -Transformed Points.32nd Conference on Neural Information Processing Systems (NeurIPS 2018)
>
> [A3] Max Jaderberg, Karen Simonyan, Andrew Zisserman, Koray Kavukcuoglu. Spatial Transformer Networks. Advances in Neural Information Processing Systems 28 (NIPS 2015)

---

> > ### Author Response · Authors · 2020-11-20
> > **Response to Reviewer1: Part Two**
> >
> > $\textbf{Q4:}$ For Section 3.2 - 3.5, the reviewer finds it very hard to follow the proposed approaches for dealing with each symmetry. (Perhaps this is because the reviewer has limited domain background in equivariant CNN) In general, the reviewer has it very hard to follow and interpret the proposed methods for dealing with each symmetry, and there is very little background provided.
> >
> > $\textbf{Answer:}$
> > We have focused on how our methods differ from well-established methods in order to work on physical data.  We will include additional background on these methods in the text and appendix.
> >
> > In summary, we employ 3 different techniques for creating equivariant networks.  These methods vary because the type of symmetry we are enforcing varies.
> >
> > i)  For uniform motion equivariance (and magnitude), we use the conjugated canonicalization described above (Answer 1) and in Section 3.4 and Section 3.5.  Our method for addressing this symmetry is novel.   As detailed in Sec 3.4 and Appendix A.3, existing methods such as Steerable CNN [10] or G-convolution [9] will not work for this symmetry group since it is non-compact and affine.
> > ii) For scale-equivariance, our method may be classified as a G-convolution [9,51].  However, as described in Answer 1 above, our method contains significant novelty to adapt to physical symmetries.
> > iii) For rotational equivariance, we use Steerable CNN [10] and specifically the e2cnn framework [49].  Here our contribution is limited to designing models within the e2cnn framework which perform well for our task.
> >
> > $\textbf{Q5:}$ It would be better if the author can provide more numerical results on real data other than simulated data. In Table 3, it seems that some of the methods with symmetry consideration are even worse than vanilla ResNet and U-net. This is a bit confusing.
> >
> > $\textbf{Answer:}$
> > the ocean dataset uses real data.  Almost every model improves on the baseline with the exception the magnitude-equivariant model on energy spectrum error.  The magnitude-equivariant model uses a weaker inductive bias than the scale-equivariant model.  It requires the model to infer the discretization step of the data.  If the model fails to do so well, this may lead to higher energy errors as we see here.

---

### Decision · Program_Chairs · 2021-01-07
**Final Decision**

**Decision:**

Accept (Poster)

**Comment:**

Symmetries play an important role in physics, and more and more papers show that they also play an important role in statistical machine learning. In particular, employing symmetries might be the key to improve training and predictive performance of machine learning models.  In this context, the present paper shows how previous physical knowledge can be leveraged to improve neural network performance, in particular within Deep dynamic models. To this end, they show how to incorporate equivariance into resnets and u-nets for dynamical systems. On a technical level, as pointed out by the reviews and also clearly mentioned by the authors, the basic building blocks are well known in the literature. However, dynamical systems also raises their own challenges resp. laws when it comes to modelling symmetries, as the authors argue in the paper and also clarified in the rebuttal. For instance, it pays off to adapt the techniques known from the literature deal better with scale, magnitude and uniform motion equivariance. This is a solid contributions and will help many other who want to apply DNNs to dynamic and physical models.